# Formin-like 1 mediates effector T cell trafficking to inflammatory sites to enable T cell-mediated autoimmunity

Scott B Thompson[1,2], Adam M Sandor[1,2], Victor Lui[1,2], Jeffrey W Chung[1,2,3], Monique M Waldman[1,2,3], Robert A Long[1,2], Miriam L Estin[1,2], Jennifer L Matsuda[1,2], Rachel S Friedman[1,2,3], Jordan Jacobelli[1,2,3]*

[1]Department of Biomedical Research, National Jewish Health, Denver, United States; [2]Department of Immunology and Microbiology, University of Colorado School of Medicine, Aurora, United States; [3]Barbara Davis Center, University of Colorado School of Medicine, Aurora, United States

**Abstract** Lymphocyte migration is essential for the function of the adaptive immune system, and regulation of T cell entry into tissues is an effective therapy in autoimmune diseases. Little is known about the specific role of cytoskeletal effectors that mediate mechanical forces and morphological changes essential for migration in complex environments. We developed a new Formin-like-1 (FMNL1) knock-out mouse model and determined that the cytoskeletal effector FMNL1 is selectively required for effector T cell trafficking to inflamed tissues, without affecting naïve T cell entry into secondary lymphoid organs. Here, we identify a FMNL1-dependent mechanism of actin polymerization at the back of the cell that enables migration of the rigid lymphocyte nucleus through restrictive barriers. Furthermore, FMNL1-deficiency impairs the ability of self-reactive effector T cells to induce autoimmune disease. Overall, our data suggest that FMNL1 may be a potential therapeutic target to specifically modulate T cell trafficking to inflammatory sites.

*For correspondence:
jordan.jacobelli@cuanschutz.edu

**Competing interests:** The authors declare that no competing interests exist.

## Introduction

T cell trafficking and extravasation into tissues are required for proper immune surveillance and execution of effector functions against pathogens and malignant cells (*Masopust and Schenkel, 2013*). However, in the context of aberrant immune function, T cell entry into target tissues can be a driver of autoimmune and inflammatory diseases (*Sallusto et al., 2012*; *Magnuson et al., 2015*; *Mellado et al., 2015*). Indeed, disruption of T cell trafficking by targeting integrins or chemotactic stimuli appears to be a promising approach to achieving immune regulation in some disease contexts (*Engelhardt and Kappos, 2008*; *Luster et al., 2005*). Many of the adhesion molecules and chemokines that orchestrate T cell trafficking have been well characterized (*Nourshargh and Alon, 2014*). However, less is known about downstream effectors that directly mediate the cytoskeletal changes necessary to elicit the force generation and shape changes inherent to the migratory process and entry into tissues.

To exit blood vessels and enter tissues, T cells undergo the multistep process of transendothelial migration (TEM). T cells are initially captured from the blood flow to begin selectin-mediated rolling along the vascular endothelial wall, followed by firm adherence in a chemokine driven integrin-mediated process. After crawling along the endothelium probing for permissive sites for extravasation, T cells initiate the diapedesis step and squeeze through the vascular endothelium (*Nourshargh and Alon, 2014*). To undergo the stages of TEM, T cells rely on morphological changes and force generation mediated by remodeling of the actin cytoskeleton (*Alon and Shulman, 2011*; *Dupré et al., 2015*; *Ridley, 2011*). Furthermore, deformation of the rigid lymphocyte nucleus is a rate-limiting

step for migration through confined environments such as during TEM (*Friedl et al., 2011*; *Alon and van Buul, 2017*). However, the contribution of specific downstream cytoskeletal effectors to the extravasation process and in mediating nuclear deformation during migration has only recently begun to be elucidated.

Actin networks in T cells consist of branched and linear filaments that are cross-linked by the motor protein non-muscle Myosin IIA (MyoIIA) (*Dupré et al., 2015*). We have previously shown that MyoIIA mediates squeezing of the nucleus during the diapedesis step of TEM (*Jacobelli et al., 2013*; *Wigton et al., 2016*). Branched actin networks, which are generated by the Arp2/3 complex and its upstream regulator WASP, play a role in T cell trafficking as well as membrane protrusion generation during TEM (*Snapper et al., 2005*). Linear actin polymerization is mediated by two major families in T cells: the Ena/VASP (vasodilator-stimulated phosphoprotein) family and the formin family. The Ena/VASP family elongates actin filaments through anti-capping activity and recruitment of actin monomers (*Chesarone and Goode, 2009*). We have recently shown that Ena/VASP proteins modulate α4 integrin expression and function in T cells to promote diapedesis (*Estin et al., 2017*). Formins remodel the actin cytoskeleton by both nucleating new actin filaments and processively elongating them through recruitment of profilin-bound actin monomers and protection from anti-capping proteins (*Faix and Grosse, 2006*). Two members of the formin family are highly expressed in T cells: Diaphanous-related formin-1 (mDia1, Diaph1) and Formin-like 1 (FMNL1, FRL1) (*Gomez et al., 2007*). Previous studies have identified a role for mDia1 in promoting T cell trafficking out of the thymus and into peripheral tissues (*Sakata et al., 2007*; *Eisenmann et al., 2007*; *Vicente-Manzanares et al., 2003*). While a specific role for mDia1 in T cell TEM has not been identified, we have recently shown that mDia1 promotes completion of diapedesis in B cell leukemia (*Thompson et al., 2018*). Unlike mDia1, which is expressed in a wide variety of cell types, FMNL1 expression is largely restricted to cells of the hematopoietic lineage (*Yayoshi-Yamamoto et al., 2000*; *Gardberg et al., 2014*; *Yue et al., 2014*). FMNL1 has previously been implicated in macrophage phagocytosis, membrane protrusion formation and migration (*Miller et al., 2017*; *Yayoshi-Yamamoto et al., 2000*; *Seth et al., 2006*; *Naj et al., 2013*). In leukemia cells, FMNL1 has been suggested to be involved in membrane dynamics, proliferation and migration (*Favaro et al., 2013*; *Han et al., 2009*; *Han et al., 2013*). However, FMNL1 is not well characterized in T cells. Previous studies have suggested that FMNL1 contributes to the reorientation of the centrosome to the immune synapse and Golgi apparatus structure in Jurkat transformed T cells as well as the cytotoxic behavior of CD8 T cells (*Gomez et al., 2007*; *Colón-Franco et al., 2011*). Additionally, FMNL1 expression has been reported to be transcriptionally upregulated by T cells in autoimmune contexts (*Odoardi et al., 2012*; *Degroote et al., 2017*). However, the role and potential mechanism of action of FMNL1 in the motility, trafficking and TEM of primary lymphocytes is unknown.

Since previous studies relied on RNAi approaches (*Gomez et al., 2007*; *Seth et al., 2006*; *Favaro et al., 2013*; *Han et al., 2009*; *Naj et al., 2013*) or myeloid specific deletion to investigate FMNL1 (*Miller et al., 2017*), we developed a novel germline FMNL1 knockout (KO) mouse to investigate the requirement and mechanism of action of FMNL1 in primary lymphocyte migration. Here we report that while FMNL1 is dispensable for lymphocyte development and homeostatic naive T cell trafficking, it is important for activated T cell trafficking to inflammatory sites and for enabling T cell-mediated autoimmunity. Investigating the mechanism of the trafficking defect, we determined that FMNL1 deficiency impairs T cell TEM at the diapedesis step due to impaired actin network remodeling and transmigration of the nucleus across endothelial barriers. Furthermore, our data show that the requirement for FMNL1 in T cell migration is dependent on barrier restrictiveness, particularly in conditions that require nucleus squeezing and deformation through small openings. Finally, we determined that FMNL1 mediates actin polymerization at the back of the T cell to promote nucleus squeezing through confined environments and that the amount of lymphocyte migration through these barriers is dictated by the level of FMNL1 expression. Together, our data identify FMNL1 as a novel mediator of T cell trafficking and transmigration across restrictive endothelial barriers and a regulator of self-reactive T cell autoimmunity.

## Results

### Deletion of FMNL1 does not significantly alter major immune cell populations

To determine the role of FMNL1 in immune cell function, we generated FMNL1 KO mice by inserting a LacZ-neomycin cassette into the *Fmnl1* locus of C57BL/6 mice (*Figure 1A*). We confirmed the correct insertion of this cassette via PCR (*Figure 1—figure supplement 1*) and verified deletion of FMNL1 at the protein level via western blot (*Figure 1B*). We then examined whether FMNL1 deficiency altered the number and frequency of major immune cell populations in the thymus, blood and secondary lymphoid organs. Flow cytometric analysis of the blood and primary and secondary lymphoid organs revealed no major differences in the number and proportions of either lymphoid (*Figure 1—figure supplement 2*) or myeloid populations (*Figure 1—figure supplement 3*) between FMNL1 KO and age matched control mice.

### FMNL1 deficiency impairs trafficking to sites of inflammation and T cell driven autoimmune disease

Having seen equivalent FMNL1 KO T cell numbers in secondary lymphoid organs, we next investigated whether FMNL1 deficiency affected activated T cell trafficking to sites of inflammation. Activated T cells are distinct from naive T cells in size, expression of adhesion molecules and chemokine receptors, and trafficking patterns. Furthermore, while the vasculature of secondary lymphoid organs is permissive for lymphocyte entry (e.g. high endothelial venules, HEV) (*Miyasaka and Tanaka, 2004*; *Bajénoff et al., 2008*), the vasculature in other tissues and organs poses a more stringent barrier to extravasation. For example, extravasation into the pancreatic islets of Langerhans during type 1 diabetes can take over an hour (*Sandor et al., 2019*), whereas entry into secondary lymphoid tissues occurs within minutes (*Bajénoff et al., 2008*). Therefore, we hypothesized that the requirements for FMNL1 in activated T cell trafficking might be different than in naive T cells.

We first examined effector T cell trafficking to the pancreatic islets in the RIP-mOVA inducible model of type 1 diabetes, in which membrane bound Ovalbumin (mOVA) is expressed in beta cells under the rat insulin promoter (RIP). To avoid any potential differences in their activation we stimulated wild-type (WT) and FMNL1 KO T cells with a strong, antigen presenting cell (APC)-independent, stimulus ex vivo using anti-CD3 and anti-CD28 antibodies. We found no major differences in the proliferation, the expression of canonical markers of activation and chemokine receptors, or the size of FMNL1 KO T cells compared to WT T cells (*Figure 2—figure supplement 1*).

To establish specific inflammation in the islets, we transferred ovalbumin specific WT OT-I transgenic T cells into RIP-mOVA mice. Six days later, we differentially fluorescently-labelled ex vivo activated polyclonal WT and FMNL1 KO CD8 T cells and co-transferred them into these RIP-mOVA

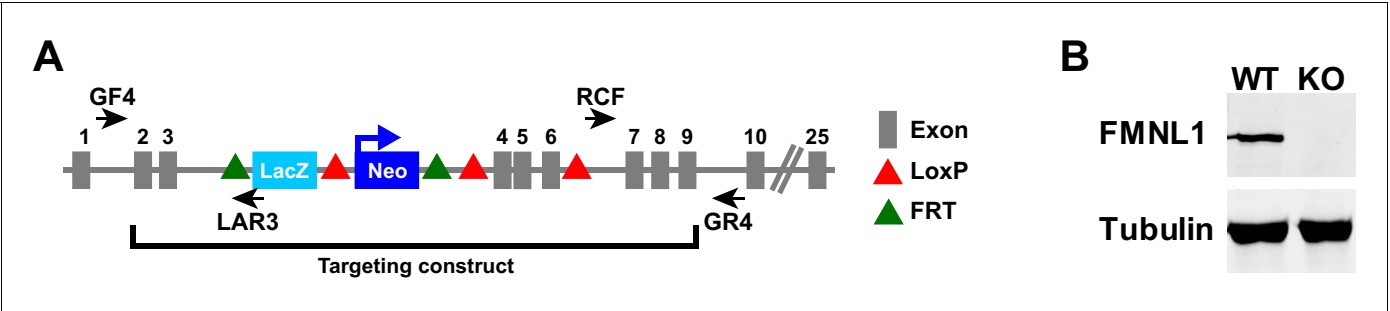

**Figure 1.** *Fmnl1* targeting strategy and knock-out confirmation. (**A**) Schematic of the targeted *Fmnl1* allele after homologous recombination with the targeting construct. (**B**) Representative western blot showing complete loss of FMNL1 protein expression in T cells from mice homozygous for the *Fmnl1* KO allele. Tubulin staining is shown as a loading control.

The online version of this article includes the following figure supplement(s) for figure 1:

**Figure supplement 1.** PCR confirmation of targeting vector insertion.

**Figure supplement 2.** T cell development and lymphocyte populations in peripheral lymphoid organs are not altered in FMNL1 KO Mice.

**Figure supplement 3.** Myeloid populations in peripheral lymphoid organs are not altered in FMNL1 KO mice.

recipient mice (*Figure 2A*). We then isolated the blood and islets 24 hr after transfer. Activated FMNL1 KO T cells displayed a 3.8-fold average reduction in trafficking to the islets compared to control T cells. (*Figure 2B,C*). Since type 1 diabetes is driven by destruction of beta cells by islet-infiltrating antigen specific T cells, we next investigated whether the observed trafficking defects in FMNL1-deficient T cells would affect their ability to induce diabetes. To test this, we activated control or FMNL1 KO OT-I and OT-II T cells ex vivo with their cognate peptides and then transferred these cells into RIP-mOVA recipients (*Figure 2D*). Recipient mice were then monitored for 28 days for signs of diabetes (2 consecutive blood glucose readings > 350 mg/dL). While 100% of mice receiving

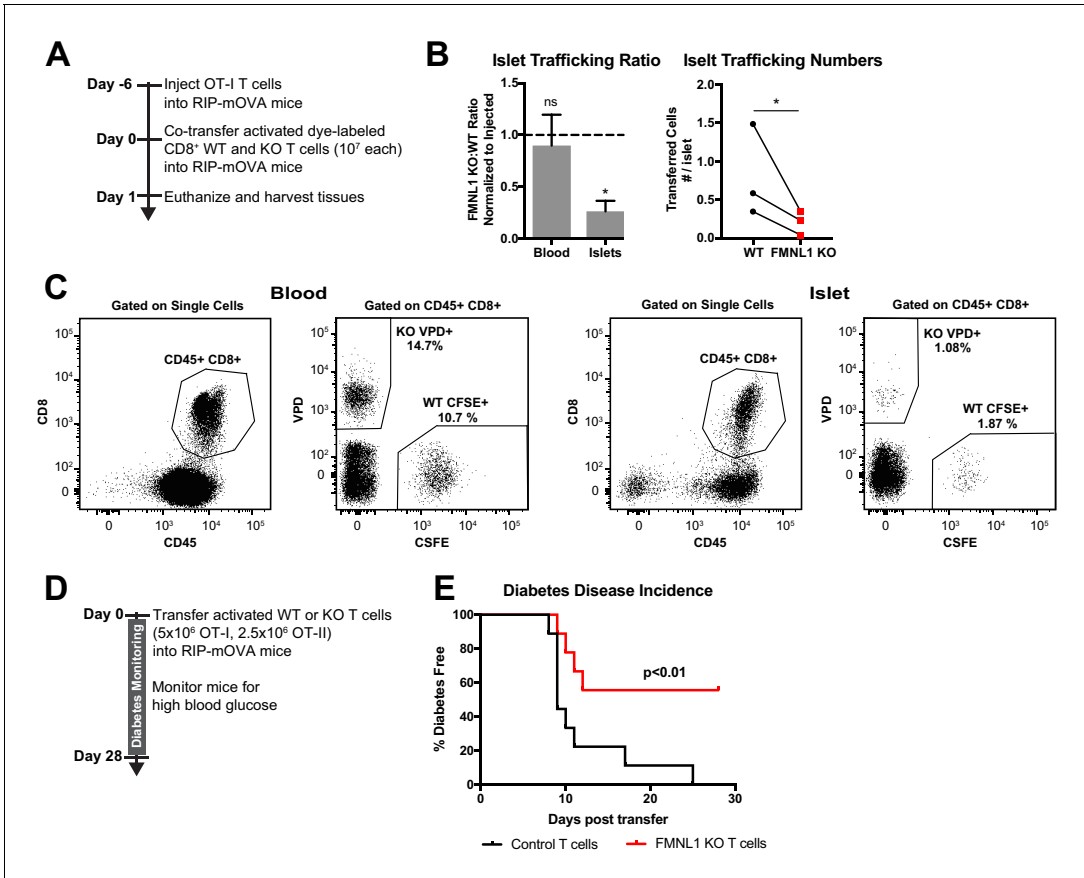

**Figure 2.** FMNL1-deficient T cells are impaired in trafficking to the pancreatic islets and inducing type 1 diabetes. (**A**) Experimental set-up to quantify activated T cell trafficking to the inflamed pancreatic islets. To induce inflammation of the pancreatic islets, 7 days prior to harvest, WT OT-I transgenic T cells were transferred intravenously into RIP-mOVA mice. Ex vivo activated, differentially dye-labeled, polyclonal CD8+ WT and FMNL1 KO T cells were then co-transferred intravenously at a 1:1 ratio into these RIP-mOVA mice and T cell trafficking to the indicated tissues was quantified by flow cytometry 24 hr post-transfer. (**B**) FMNL1-deficient T cells are impaired in trafficking to the inflamed pancreatic islets. Ratio of co-transferred FMNL1 KO to WT T cells in the indicated tissues (left). Values were normalized to the FMNL1 KO:WT ratio in the injected sample. A ratio below 1.0 (dashed line) indicates impaired trafficking to the tissue. Number of transferred WT or FMNL1 KO T cells recovered from the islets (right). (**C**) Representative blood and pancreatic islet flow plots and gating strategy for identifying co-transferred dye-labeled T cells. VPD = Violet Proliferation Dye, CFSE = Carboxyfluorescein succinimidyl ester. (**D**) Experimental set up for induction of type 1 diabetes in RIP-mOVA mice. Control or FMNL1 KO OT-I and OT-II T cells were activated ex vivo and then transferred into RIP-mOVA recipients. Recipient mice were then monitored for 28 days for glycemia. (**E**) FMNL1 deficiency in self-reactive T cells delays and partially protects from diabetes. Diabetes incidence of mice receiving control or FMNL1 KO OT-I and OT-II T cells. Data in B are the mean ± SEM (left) or individual means (right) from 3 independent experiments with at least 3 recipient mice per experiment; data in E are pooled from 3 independent experiments with cohorts of 3 mice/group each. Statistics in B were calculated using a one-sample two-tailed t-test against a theoretical FMNL1 KO:WT ratio of 1.0 (left) or a two-tailed paired t test (right); statistics in E were calculated by Log-rank test. n.s. = not significant, *=p < 0.05.

The online version of this article includes the following figure supplement(s) for figure 2:

**Figure supplement 1.** FMNL1 deficiency does not impair ex vivo T cell activation.

control T cells developed diabetes, only 44% of mice receiving FMNL1 KO T cells developed diabetes (*Figure 2E*).

We next wanted to determine whether our observation on FMNL1 regulation of T cell trafficking to the islets extended to other inflamed tissues and autoimmune disease settings. The central nervous system (CNS) also has highly restrictive vascular endothelial barriers. Direct observation of extravasation into the CNS suggests that T cells can take several hours to access the parenchyma of the brain and spinal cord (*Vajkoczy et al., 2001*). We therefore sought to examine how FMNL1 deficiency would affect CD4 T cell trafficking to the CNS in the experimental autoimmune encephalitis (EAE) model of CNS inflammation. We used a standard model of EAE induction by immunizing with myelin oligodendrocyte glycoprotein (MOG) peptide in Complete Freund's Adjuvant (CFA). After development of overt EAE symptoms in recipient mice (disease score ≥2), we co-transferred differentially dye-labled ex vivo activated polyclonal WT and FMNL1 KO CD4 T cells (*Figure 3A*). Twenty-four hours post-transfer, we quantified the number of transferred cells in the blood, brain, and spinal cord of the recipient mice. Activated FMNL1 KO T cells had a 1.4-fold reduction in trafficking to the brain and a 2.2-fold reduction in trafficking to the spinal cord compared to control T cells (*Figure 3B,C*).

Transfer of MOG specific 2D2 T cells can drive EAE in recipient mice in a manner that relies on T cell trafficking to the CNS (*Jäger et al., 2009*). Therefore, we examined whether the observed trafficking defects of FMNL1-deficient T cells would affect their ability to induce EAE in WT recipient mice. We activated control or FMNL1 KO 2D2 CD4 T cells with MOG peptide ex vivo and confirmed that ex-vivo stimulation with cognate self-peptide yielded similar expression of canonical markers of activation and proliferation of FMNL1 KO T cells compared to control T cells (*Figure 3—figure supplement 1*). We then transferred these activated T cells into recipient mice, and monitored these mice for development of EAE (*Figure 3D*). While 100% of mice receiving control 2D2 T cells developed EAE, only 33% of mice receiving FMNL1 KO 2D2 T cells developed EAE (*Figure 3E*). Mice receiving control T cells displayed an average peak disease score of 3.5 compared to an average peak score of 0.67 for mice receiving FMNL1 KO T cells (*Figure 3E*). Given that similar trafficking and disease induction defects were observed in two different autoimmune models, together our data support that FMNL1 broadly promotes trafficking of effector T cells to sites of inflammation enabling them to cause autoimmune disease.

## FMNL1 deficiency does not affect T cell trafficking to secondary lymphoid organs

Having seen impaired trafficking of FMNL1 KO T cells to peripheral inflamed tissues, we wanted to determine whether the impairment in FMNL1-deficient T cell trafficking was specific to restrictive non-lymphoid tissues or would also apply to more permissive lymphoid tissues. Additionally, we wanted to investigate how T cell activation status affected this process. First, we examined naïve T cell trafficking to secondary lymphoid organs under homeostatic conditions. We used co-adoptive transfer of differentially fluorescently-labeled naive CD45.2 control and FMNL1 KO T cells into CD45.1 congenic WT recipient mice (*Figure 3—figure supplement 2A*). Twenty-four hours post-intravenous transfer, we found equivalent ratios and numbers of transferred control and FMNL1 KO naïve T cells in the spleen, lymph nodes, and blood of recipient mice (*Figure 3—figure supplement 2B,C*).

Next, we examined if activated FMNL1 KO T cell trafficking to secondary lymphoid organs was impaired. We co-transferred differentially dye-labeled ex vivo activated CD45.2 WT and FMNL1 KO T cells into CD45.1 congenic WT recipient mice (*Figure 3—figure supplement 2D*). Twenty-four hours later we quantified the number of transferred WT and FMNL1 KO T cells in the blood, spleen, and lymph nodes of recipient mice. The ratio of FMNL1 KO and WT T cells in lymph nodes was equivalent, while the ratio of KO to WT T cells was slightly lower than the expected value of 1 in the blood and greater than the expected value of 1 in the spleen. However, comparison of the number of cells present in these locations revealed no significant differences (*Figure 3—figure supplement 2E*). Taken together, our data suggest that FMNL1 is dispensable for homeostatic T cell trafficking to permissive secondary lymphoid organs but relied upon for trafficking to more restrictive inflamed tissues.

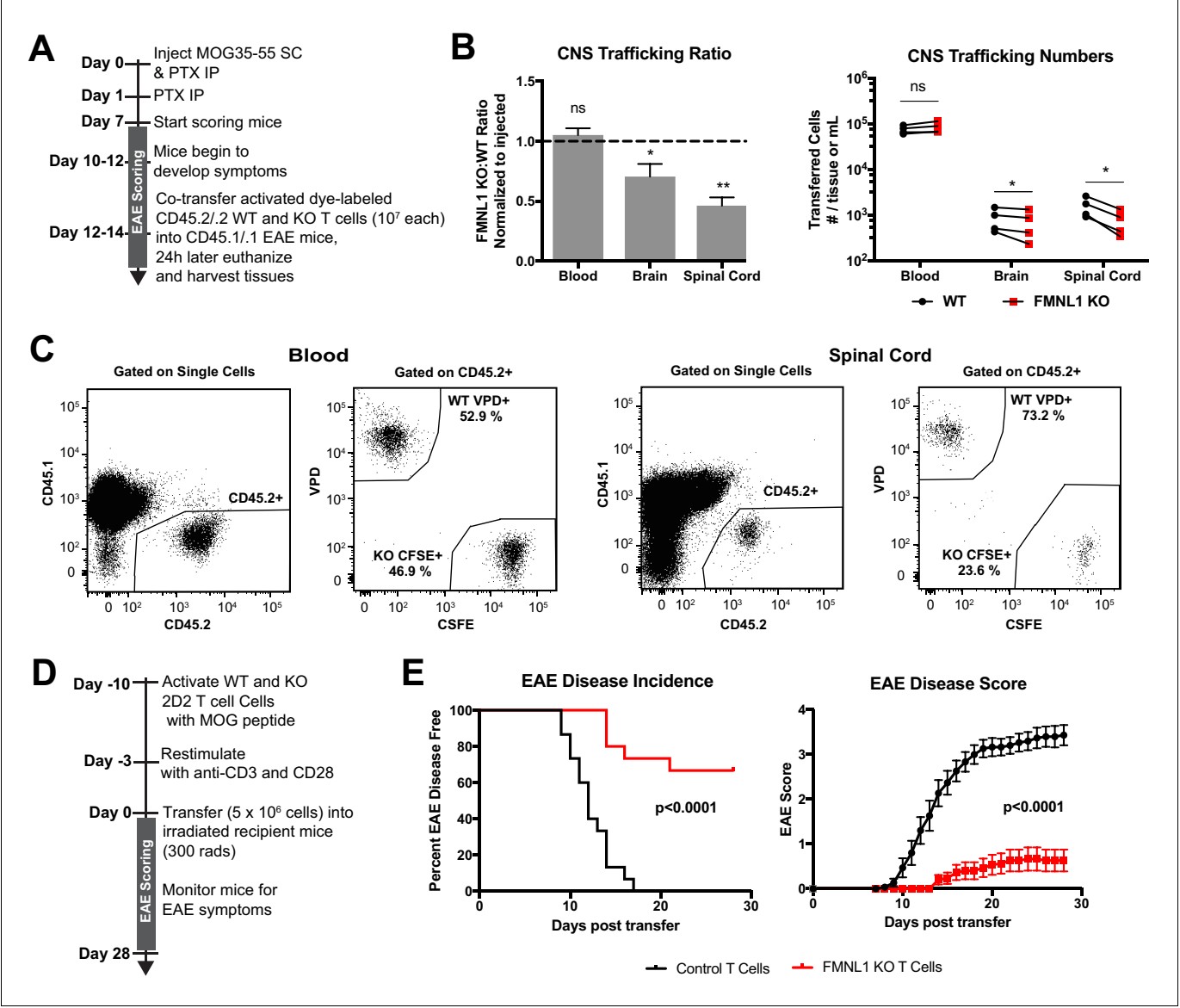

**Figure 3.** FMNL1-deficient T cells are impaired in trafficking to the CNS and inducing EAE. (**A**) Experimental set-up to quantify activated T cell trafficking to the inflamed CNS. Activated, differentially dye-labeled, polyclonal CD45.2/.2 WT and FMNL1 KO T cells were co-transferred intravenously at a 1:1 ratio into CD45.1/.1 mice with ongoing EAE (score ≥2.0) and T cell trafficking to the indicated tissues was quantified by flow cytometry 24 hr post-transfer. (**B**) FMNL1-deficient T cells are impaired in trafficking to the inflamed CNS. Ratio of co-transferred FMNL1 KO to WT T cells in the indicated tissues (left). Values were normalized to the FMNL1 KO:WT ratio in the injected sample. A ratio below 1.0 (dashed line) indicates impaired trafficking to the tissue. Number of transferred WT or FMNL1 KO T cells recovered from the indicated tissues (right). (**C**) Representative spinal cord and blood flow plots and gating strategy for identifying co-transferred dye-labeled T cells quantified in B. (**D**) Experimental set up for induction of EAE via T cell transfer. Control or FMNL1 KO MOG-specific 2D2 T cells were activated ex vivo and then transferred into WT recipient mice. EAE disease severity was scored daily for 28 days. (**E**) FMNL1 deficiency in T cells delays EAE onset and partially protects from disease. EAE incidence (score ≥1.0) in mice receiving control or FMNLL1 KO 2D2 T cells (left). Mean EAE score ± SEM over time in mice receiving control or FMNL1 T cells 2D2 T cells (right). Data in B are the mean ± SEM (left) or individual means (right) from 4 independent experiments with 2 recipient mice per group; data in E are pooled from 3 independent experiments with cohorts of 5 mice/group each. Statistics in B were calculated using a one-sample two-tailed t-test against a theoretical FMNL1 KO:WT ratio of 1.0 (left) or repeated measures one-way ANOVA with Sidak's multiple comparisons test (right); statistics in E (left) were calculated by Log-rank test; statistical interaction of genotype with disease severity over time in E (right) was calculated by repeated measures two-way ANOVA. n.s. = not significant, *=p < 0.05, **=p < 0.01.

The online version of this article includes the following figure supplement(s) for figure 3:

**Figure supplement 1.** FMNL1 deficiency does not impair ex vivo T cell cognate peptide stimulation.

**Figure supplement 2.** FMNL1 deficiency does not impair T cell trafficking to lymphoid tissues.

## FMNL1 deficiency impairs T cell TEM at the diapedesis step

Having established a role for FMNL1 in activated T cell trafficking, to determine how FMNL1 regulates this process, we next investigated which stages of TEM (rolling, adhering, crawling, and diapedesis), were affected by FMNL1 deficiency. For these experiments, we employed a previously established in vitro flow chamber system to visualize and quantify T cell TEM under shear flow using time-lapse microscopy (*Jacobelli et al., 2013*; *Wigton et al., 2016*; *Estin et al., 2017*; *Thompson et al., 2018*). After culturing an endothelial cell monolayer within the flow chamber, we perfused in differentially dye-labeled activated control and FMNL1 KO T cells and imaged them under physiological shear flow for 30 min. With phase-contrast imaging, T cells above the plane of the endothelium display a white halo, which is progressively lost as the cells undergo diapedesis (*Figure 4A*, *Video 1*). Cells that lose a portion of the white phase halo localized to a small protrusion of the cell, detected by fluorescence, are considered to have initiated diapedesis (*Figure 4A,B*, and *Videos 1* and *2*). Completion of diapedesis is scored as complete loss of the phase halo. Quantification of the number of cells adhering to the endothelium and subsequently detaching under flow, revealed no differences between WT and FMNL1 KO T cells (*Figure 4C,D*). Similarly, both WT and FMNL1 KO T cells were equally capable of crawling on the endothelial monolayer (*Figure 4E*), suggesting that FMNL1 is dispensable during the first stages of TEM. While FMNL1 KO T cells were equivalent to WT cells in their ability to initiate diapedesis (*Figure 4F*), they had a 3.8-fold reduction in their ability to complete the TEM process (*Figure 4G*). Furthermore, the few FMNL1 KO T cells that were able to complete TEM took more than twice as long on average compared to WT T cells (*Figure 4H*). These data indicate a specific role for FMNL1 in completion of the diapedesis step of the TEM process.

## FMNL1 deficiency impairs transmigration of the nucleus during TEM

To further elucidate how FMNL1 promotes T cell diapedesis, we examined its localization within T cells during TEM. Using the same flow chamber system as above, dye-labeled activated T cells were introduced into the flow chamber and allowed to migrate on the endothelial monolayer under shear flow for 5 min before fixing the cells. We then stained for FMNL1 and the nucleus (*Figure 5A*). We confirmed that the FMNL1 antibody staining was specific for FMNL1 using similar staining conditions on control and FMNL1 KO T cells (*Figure 5—figure supplement 1*). Using the plane of the endothelium as a reference, we identified cells that were actively undergoing diapedesis, i.e. with a portion of their cytoplasm below the endothelial monolayer (*Figure 5B*). In these cells, we then quantified the fluorescence intensity of the nucleus, FMNL1, and cytoplasmic staining from the back to the front of the cell (*Figure 5C*). Cells were scored as having perinuclear FMNL1 enrichment, partial perinuclear enrichment, or no enrichment (see Methods for analysis). On average we found posterior perinuclear enrichment of FMNL1 in 64% of transmigrating WT T cells, with an additional 12.5% displaying a partial perinuclear enrichment (*Figure 5D*).

The rigid structure of the nucleus can impede completion of migration through restrictive barriers such as vascular endothelium (*Friedl et al., 2011*; *Jacobelli et al., 2013*). Given the localization of FMNL1 directly behind the nucleus of transmigrating T cells, we examined whether FMNL1 deficiency would impact the ability of activated T cells to translocate their nuclei across the endothelium during TEM. Using differentially dye-labeled WT and FMNL1 KO T cells in the same flow chamber system as above, after 5 min of migration, we fixed the cells and quantified the position of the T cell nucleus relative to the endothelium. FMNL1 KO T cells were significantly impaired in their ability to transmigrate their nuclei, with only 7% of transmigrating T cells having migrated their nuclei below the endothelium compared to 32% for WT T cells (*Figure 5E*). This finding is consistent with our earlier observation that FMNL1 KO T cells are able to initiate diapedesis but are substantially impaired in completing the process. Taken together with the localization data, these findings suggest that FMNL1 facilitates completion of diapedesis by promoting the transmigration of the rigid nucleus through the endothelial barrier.

## T cell migration through narrow pores is FMNL1 dependent

Based on our observations that FMNL1 deficiency impaired activated T cell trafficking selectively to tissues with restrictive endothelial barriers and impaired TEM specifically at the diapedesis step, we hypothesized that the requirement for FMNL1 in migration would be dependent on the physical

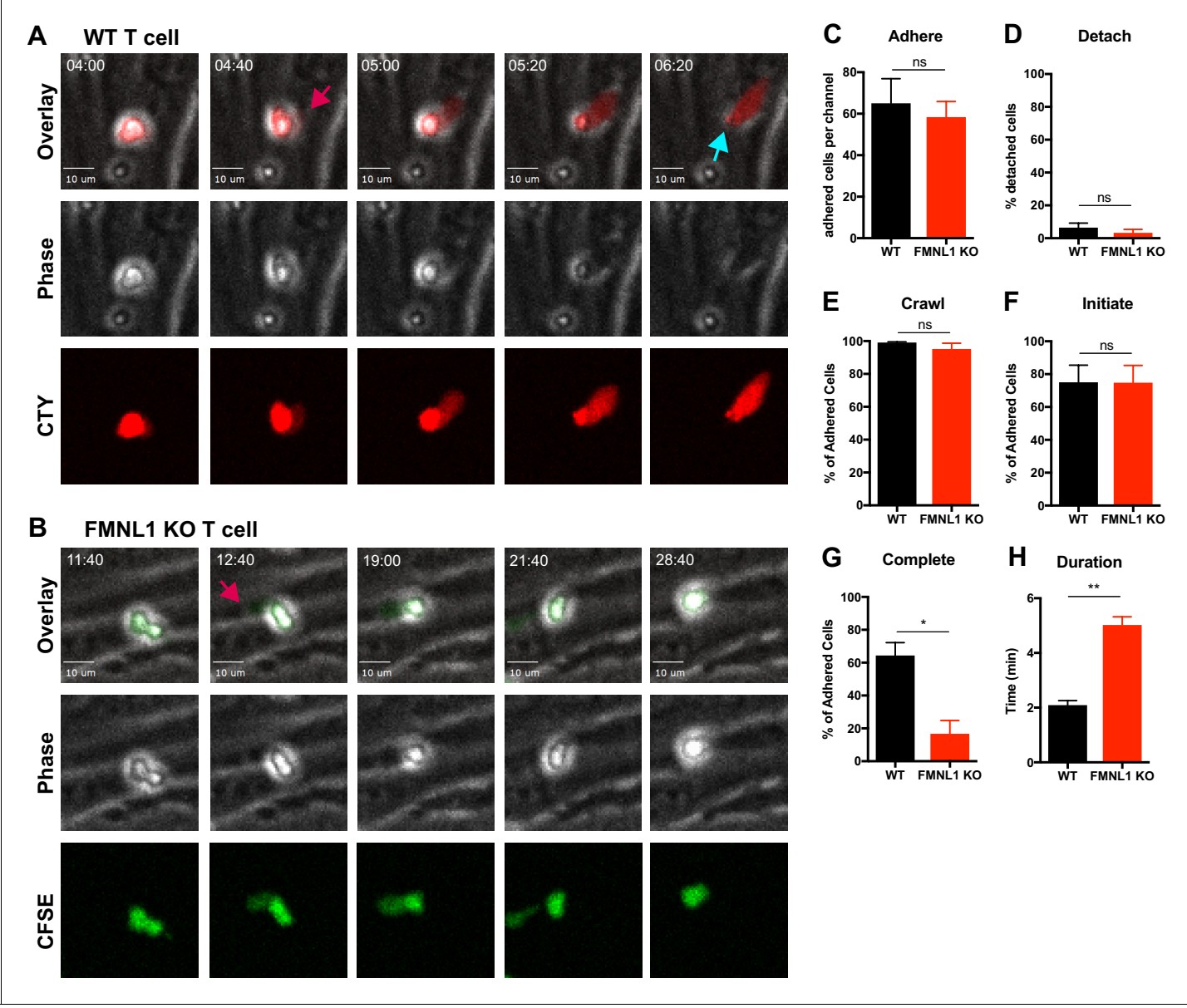

**Figure 4.** FMNL1 deficiency impairs T cell TEM at the diapedesis step. Ex vivo activated CFSE or CellTrace Yellow (CTY) dye-labeled T cells were perfused into flow chambers containing bEnd.3 brain endothelial cell monolayers and kept under shear flow (2 dyne/cm$^2$) for up to 30 min. During this time, phase contrast and fluorescence images were acquired every 20 s using a spinning-disk confocal microscope. (A) Selected time-points of a representative WT T cell during transmigration (*Video 1*). This transmigrating T cell undergoes trans-endothelial migration (TEM) evidenced by a stepwise darkening in the Phase contrast channel during the time-lapse. The red arrow points to the formation of membrane protrusions under the endothelial monolayer; the blue arrow points to the completion of TEM as shown by the disappearance of the phase halo. Time in min:s. (B) Selected time-points of a representative FMNL1 KO T cell attempting transmigration (*Video 2*). The red arrow points to the formation of membrane protrusions under the endothelial monolayer. However, this FMNL1 KO T cell never completes TEM as evidenced by the preservation of the phase halo. Time in min:s. (C) FMNL1 deficiency does not alter the ability of T cells to adhere to the endothelial monolayer. Number of T cells adhered to the endothelial monolayer. (D) FMNL1 deficiency does not affect T cell detachment from the endothelial monolayer. Percentage of adhered T cells that detached from the endothelial monolayer. (E) WT and FMNL1 KO T cells have similar crawling behavior. Percentage of adhered T cells that crawled on the endothelial monolayer. (F) FMNL1 deficiency does not impair the ability of T cells to attempt TEM. Percentage of adhered cells that attempted TEM as evidenced by extension of membrane protrusions underneath the endothelial monolayer. (G) FMNL1 deficiency strongly impairs the ability of T cells to complete TEM. Percentage of adhered cells that completed TEM as evidenced by complete loss of the phase halo. (H) FMNL1 deficiency prolongs TEM duration. For cells able to complete TEM, the time in minutes from first attempt to completion was quantified. Statistics in C-H calculated using two-tailed paired t-tests. Data in C-H are the mean ± SEM from 3 independent experiments with >100 cells analyzed per experiment. n.s. = not significant, *=p < 0.05, **=p < 0.01.

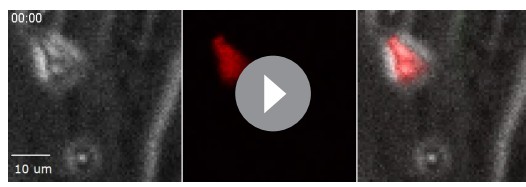

**Video 1.** Example of a WT control T cell undergoing transendothelial migration imaged by time-lapse confocal microscopy. Control and FMNL1 KO T cells were activated ex-vivo and differentially dye-labeled with CFSE or CellTrace Yellow (CTY). These T cells were mixed at a 1:1 ratio and perfused into flow chambers containing bEnd.3 brain endothelial cell monolayers and kept under shear flow (2 dyne/cm$^2$) for up to 30 min. During this time, phase contrast and fluorescence images were acquired every 20 s using a spinning-disk confocal microscope. Phase contrast (left), CTY fluorescence (middle), and overlay (right) images are shown. This representative control T cell completes transendothelial migration as shown by the progressive disappearance of the white phase contrast ring around the T cell. Time is displayed as min:s.
https://elifesciences.org/articles/58046#video1

restriction of the barrier. To test this idea, we performed transwell assays with inserts containing pore sizes of different diameters. In response to CXCL12, activated FMNL1 KO cells were impaired in their ability to migrate across 3 µm transwell inserts compared to activated WT T cells (*Figure 6A*). However, FMNL1 KO cells migrated to CXCL12 equivalently as WT T cells across 5 µm transwell inserts (*Figure 6A*). This effect was not exclusive to CXCL12, in fact in response to CXCL10, FMNL1 KO T cells were similarly impaired in migrating across 3 µm, but not 5 µm transwell inserts (*Figure 6—figure supplement 1A*). The nuclei of lymphocytes have been estimated to be approximately 5–7 µm in diameter (*Friedl et al., 2011*). Thus, our observation that FMNL1-deficient T cells are impaired in transmigration of their nuclei across endothelial barriers is consistent with our finding that FMNL1-deficient T cells are impaired in transmigrating through 3 µm but not 5 µm transwell pores. Neutrophils also express FMNL1 at similar levels to T cells (*Ericson et al., 2014*) but are characterized by a multi-lobed, more flexible, nuclear structure (*Feng et al., 1998*; *Mackarel et al., 1999*; *Yadav et al., 2018*). Therefore, we examined whether FMNL1-deficient neutrophils would be impaired in migrating through 3 µm transwell pores. In response to the neutrophil chemoattractant CXCL1, FMNL1 KO neutrophils displayed equivalent migration compared to WT neutrophils (*Figure 6B*). These findings suggest that in cell types with more flexible nuclei, the need for FMNL1 in transmigrating through restrictive openings is reduced.

Chemokine signaling during T cell migration can trigger actin remodeling. To further determine the mechanism by which FMNL1 regulates T cell TEM, we next examined the ability of activated FMNL1-deficient T cells to polymerize actin in response to chemokine using fluorescent phalloidin staining to quantify filamentous actin (F-actin). Unstimulated WT and FMNL1 KO T cells displayed similar baseline levels of F-actin (*Figure 6C*). In response to 15 s of CXCL12 stimulus, WT T cells increased their total polymerized actin an average of 1.8 fold, while FMNL1 KO T cell only increased their total polymerized actin an average of 1.4 fold (*Figure 6D*). We also observed a similar impairment in FMNL1 KO T cell actin polymerization in response to CXCL10 stimulus (*Figure 6—figure supplement 1B*). Thus, while FMNL1 KO T cells are capable of polymerizing actin in response to chemokines, their ability to respond is ~22% reduced compared to WT T cells. Given the presence of other actin polymerizing proteins such as Arp2/3 and mDia1, this partial rather than complete inhibition is not surprising but still highlights a contribution of FMNL1 in actin remodeling in response to chemokine signaling.

Furthermore, to quantitatively test the dependence of T cell transmigration on FMNL1

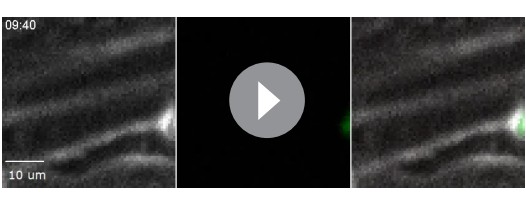

**Video 2.** Example of a FMNL1 KO T cell unable to complete transendothelial migration. Control and FMNL1 KO T cells were activated ex-vivo and differentially dye-labeled with CFSE or CellTrace Yellow (CTY). These T cells were mixed at a 1:1 ratio and perfused into flow chambers containing bEnd.3 brain endothelial cell monolayers and kept under shear flow (2 dyne/cm$^2$) for up to 30 min. During this time, phase contrast and fluorescence images were acquired every 20 s using a spinning-disk confocal microscope. Phase contrast (left), CFSE fluorescence (middle), and overlay (right) images are shown. This representative FMNL1 T cell fails to complete transendothelial migration as shown by the persistence of the white phase contrast ring around the T cell. Time is displayed as min:s.
https://elifesciences.org/articles/58046#video2

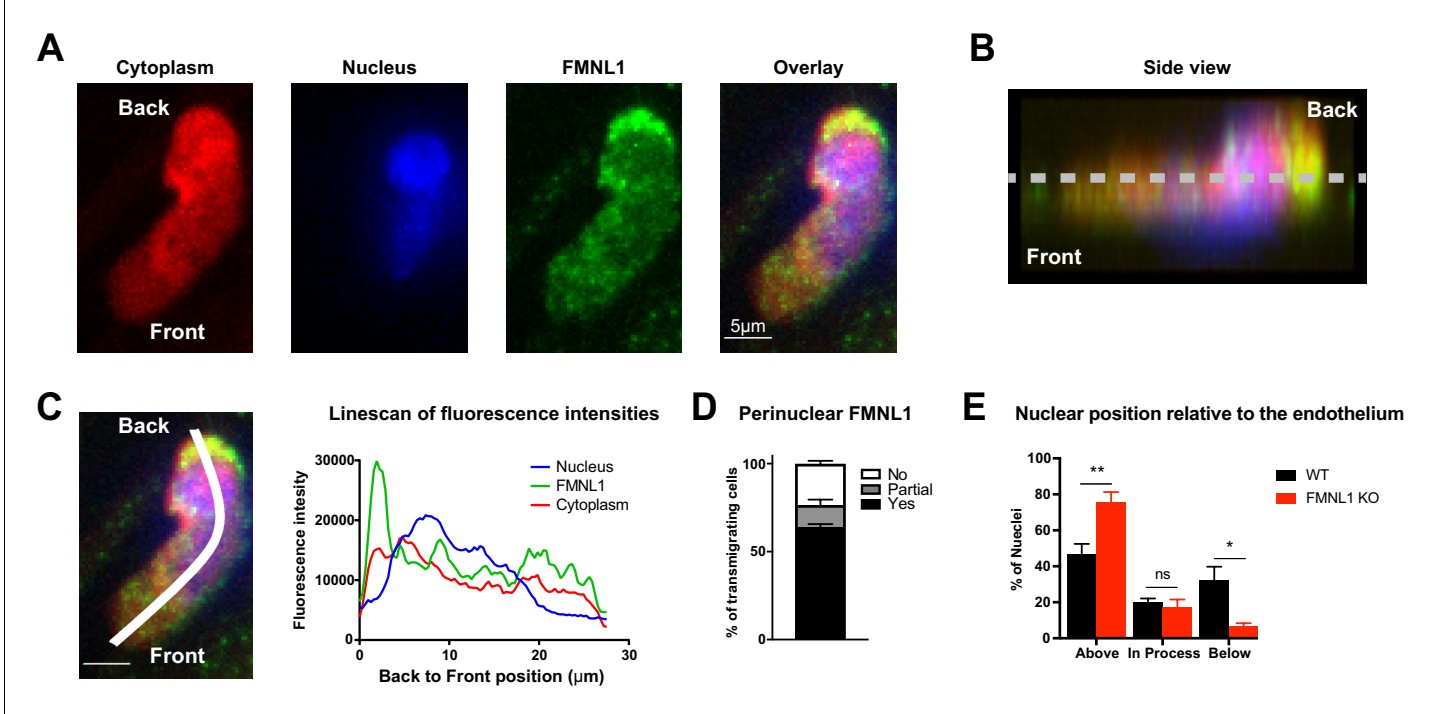

**Figure 5.** FMNL1 promotes nucleus transmigration in T cells undergoing TEM. TEM of dye-labeled WT T cells was set up as in *Figure 4*. After 5 min of TEM under flow cells were fixed, permeabilized, and stained with DAPI and an anti-FMNL1 antibody (**A**) Representative maximum Z projections of a transmigrating T cell, defined as a cell with a portion of the cytoplasm underneath the endothelial monolayer. (**B**) Representative 3D side view reconstruction of the transmigrating T cell in A. Dashed line indicates position of the endothelial monolayer. (**C**) Representative linescan quantification of fluorescence intensities in a transmigrating T cell. Graph of the fluorescence intensities along the depicted line in each channel relative to the position within the cell. (**D**) FMNL1 is enriched behind the nucleus in transmigrating T cells. Percentage of transmigrating T cells with perinuclear enrichment behind the nucleus, partial enrichment of FMNL1 behind the nucleus, or no enrichment (see Methods for analysis). (**E**) FMNL1 deficiency impairs transmigration of the T cell nucleus across endothelial barriers. Position of T cell nuclei in fixed WT or FMNL1 KO T cells after 5 min of TEM under flow. Using spinning-disk confocal microscopy and DAPI staining, the nuclei were scored as being above the plane of the endothelium, in the process of transmigrating, or below. Data in D are the mean ± SEM from 3 independent experiments with >15 cells analyzed per experiment; data in E are the mean ± SEM from 4 independent experiments with >50 cells analyzed per experiment. Statistics in E were calculated using repeated-measures ANOVA with Sidak's multiple comparisons test. n.s. = not significant, *=p < 0.05, **=p < 0.01.

The online version of this article includes the following figure supplement(s) for figure 5:

**Figure supplement 1.** Immunofluorescent FMNL1 antibody staining is specific for FMNL1.

expression, we expressed FMNL1 in activated WT or FMNL1 KO T cells using retroviral constructs. As expected, FMNL1 KO T cells receiving a control construct were impaired in 3 µm transwell migration to CXCL12 compared to WT T cells receiving the same construct (*Figure 6E*). However, re-expression of FMNL1 in FMNL1 KO T cells restored their ability to transmigrate through the 3 µm pores (*Figure 6E*). To determine if there was a correlation between FMNL1 levels and the ability of T cells to transmigrate, in parallel, we measured the level of FMNL1 expression in WT and FMNL1 KO T cells receiving either the control or FMNL1-expressing constructs using western blotting (*Figure 6—figure supplement 2*). We then compared the level of FMNL1 expression to the ability of cells from the same population to migrate in a 3 µm transwell assay. We found a strong positive correlation between the level of FMNL1 expressed by a given population of T cells and their ability to transmigrate through restrictive pores (*Figure 6F*). This finding further supports that FMNL1 regulates the ability of activated T cells to navigate restrictive barriers.

## FMNL1 mediates posterior perinuclear actin polymerization to promote T cell migration through environmental constrictions

Having determined that FMNL1 promotes extravasation by facilitating nucleus passage through the endothelial barrier, we further investigated the mechanism by which FMNL1 promotes T cell

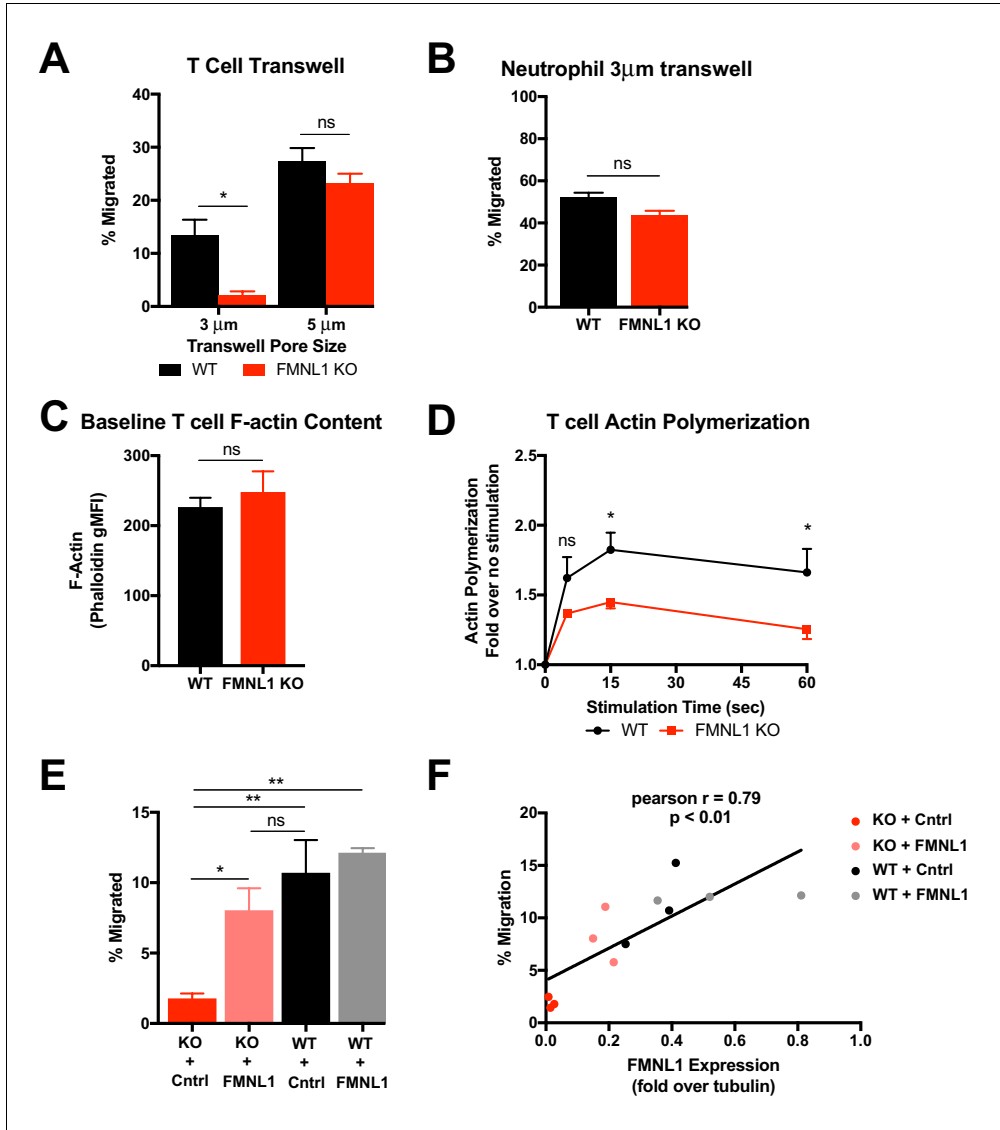

**Figure 6.** T cell migration through narrow pores is FMNL1 dependent. (**A**) FMNL1-deficient T cells are impaired in chemotaxis through narrow pores. Percentage of activated cells migrating through transwell membranes of the indicated pore size in response to CXCL12. (**B**) Neutrophil chemotaxis through narrow pores is not impaired by FMNL1 deficiency. Percentage of neutrophils migrating through 3 µm transwell membrane pores in response to CXCL1. (**C**) FMNL1-deficient T cells express similar baseline levels of F-actin compared to WT T cells. Quantification of F-actin (gMFI of fluorescent phalloidin staining by flow cytometry) in activated T cells. (**D**) FMNL1-deficient T cells are impaired in actin polymerization in response to chemokine. Time course of fold increase in F-actin (determined as above) of activated T cells in response to CXCL12 stimulation. (**E**) Re-expression of FMNL1 in FMNL1 KO T cells restores migration through narrow pores. Activated WT or FMNL1 KO T cells were transduced using either a FMNL1-expressing or control retroviral construct. Percentage of indicated T cells migrating through 3 µm transwell membrane pores in response to CXCL12. (**F**) Transwell migration correlates with FMNL1 expression. Percentage of T cells migrating through 3 µm transwell membrane pores in response to CXCL12 vs level of FMNL1 expression as determined by densitometry of western blot staining (intensity of FMNL1 staining over intensity of tubulin staining). The line indicates simple linear regression of the data. Data in A-E are the mean ± SEM from 3 independent experiments; data in F are the individual values from 3 independent experiments. Statistics in A and E were calculated by repeated measures one-way ANOVA with Sidak's multiple comparisons test; statistics in B and C were calculated using a two-tailed paired t-test; statistics in D were calculated by repeated-measures two-way ANOVA with Sidak's multiple comparisons test; statistics in F were calculated using Pearson's correlation. n.s. = not significant, *=p < 0.05, **=p < 0.01.

The online version of this article includes the following figure supplement(s) for figure 6:

*Figure 6 continued on next page*

*Figure 6 continued*

**Figure supplement 1.** Transwell migration and actin polymerization responses to CXCL10.
**Figure supplement 2.** Re-expression of FMNL1 in KO cells.

migration through restrictive environments. Our data showed that during migration through endothelial barriers FMNL1 is enriched at the back of the T cell and that FMNL1-deficient T cell nuclei are impaired in translocating through the endothelial barrier. Therefore, we employed an in vitro system that would allow us to image actin dynamics during T cell migration in a well-defined environment through unconfined and confined spaces. To this end, we used microfabricated microchannels with defined channel size and constrictions (*Figure 7A*). Using this system, we imaged control and FMNL1 KO T cell migration within these microchannels and found that, similar to our transwell system results, FMNL1-deficient T cells were significantly impaired in migrating through the 3 μm constriction points in the microchannels (*Figure 7B*). We next used T cells derived from control LifeAct-GFP (*Riedl et al., 2010*) or FMNL1 KO/LifeAct-GFP mice to determine actin dynamics during migration within the microchannels using time-lapse microscopy (*Figure 7C* and *Videos 3* and *4*). We measured the F-actin content at the front and back of T cells migrating in the unconfined portions of the microchannels and at the time their nucleus engaged a constriction. Our analysis shows that control T cells enrich their F-actin content around the back of the nucleus when it has to deform upon encountering a constriction. However, FMNL1-deficient T cells were impaired in their ability to generate this posterior perinuclear F-actin enrichment when encountering a constriction (*Figure 7D and E*). Taken together, these data support that FMNL1 mediates F-actin polymerization to promote nuclear passage through constrictions enabling efficient T cell migration under confinement.

## Discussion

Lymphocyte extravasation out of the blood stream into tissues is a critical process for proper immune function (*Masopust and Schenkel, 2013*). In the context of autoimmunity, the trafficking of self-reactive lymphocytes to the disease site is a key step in mediating autoimmune disease. The morphological changes and force generation inherent to the extravasation process are mediated by dynamic changes to the actin cytoskeleton, but the role and mechanism of action of specific cytoskeletal effectors in regulating these effects is not well understood (*Nourshargh and Alon, 2014*; *Alon and Shulman, 2011*; *Dupré et al., 2015*). In this study, we report a novel role for FMNL1 in mediating activated T cell trafficking into inflamed tissues by promoting transmigration of the nucleus through restrictive endothelial barriers. Furthermore, we found that FMNL1 deficiency impaired the ability of self-reactive T cells to induce autoimmune disease. Together, these findings identify a previously unknown FMNL1-dependent mechanism that enables T cells to enter non-lymphoid tissues to execute their effector functions, enabling autoimmune pathology.

To our knowledge, this is the first report of germline FMNL1 KO mice and the first investigation of the effects of FMNL1 deficiency in T cell migration and autoimmune disease in vivo. A prior report suggested that complete FMNL1 deficiency in mice would be embryonically lethal (*Miller et al., 2017*). However, our FMNL1-deficient mice are viable and breed normally. This discrepancy may be due to differences in the targeting approach and use of a ubiquitously expressed Cre recombinase in the earlier approach. Our observation of normal populations of immune cells within the blood, spleen, and lymph nodes of FMNL1-deficient mice suggests that FMNL1 is dispensable for hematopoiesis. It is likely that other formins, such as mDia1, have a more predominant role in lymphocyte development and proliferation, and/or can compensate for the loss of FMNL1. Thus, our observations here present the opportunity to examine the function of FMNL1 in T cell trafficking and autoimmune disease without the potential confounding effects of altered lymphocyte development and proliferation.

The remodeling of the actin cytoskeleton is critical for morphological changes and mechanotransduction throughout the TEM process (*Dupré et al., 2015*; *Alon and Shulman, 2011*; *Ridley, 2011*). Anchoring of integrins to the actin cytoskeleton is critical for firm adhesion and crawling upon to the endothelial monolayer, while the generation of actin-rich membrane protrusions is important for probing the endothelial monolayer and initiating diapedesis (*Alon and Shulman, 2011*;

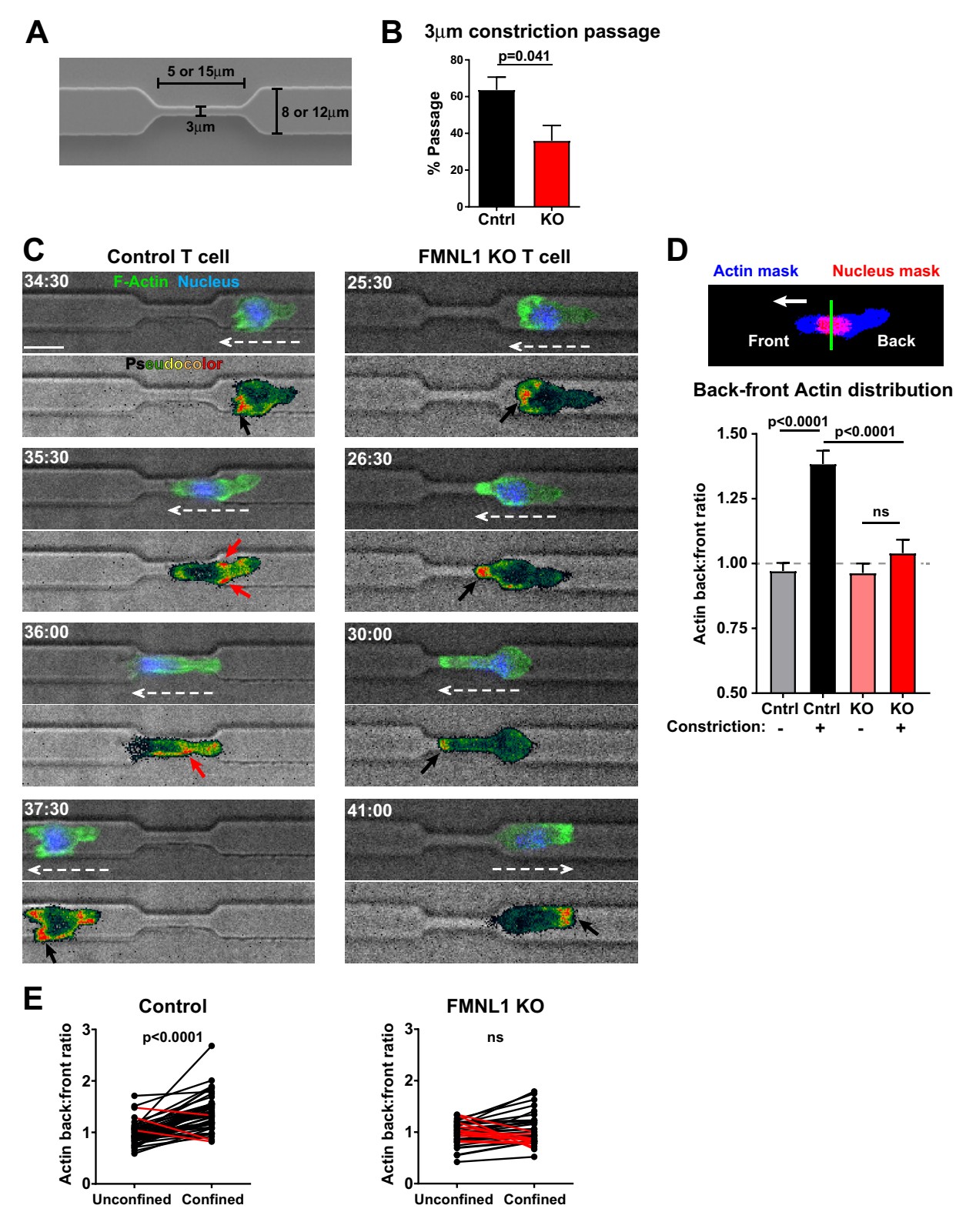

**Figure 7.** FMNL1 mediates posterior perinuclear actin polymerization to promote T cell migration through environmental constrictions. Ex vivo activated control LifeAct-GFP and FMNL1 KO LifeAct-GFP T cells were stained with Hoechst, added to PDMS microchannels and imaged by spinning-disk confocal time-lapse microscopy. (**A**) Schematic of the PDMS microchannels with constrictions used in the experiments. (**B**) FMNL1 deficiency impairs the ability of T cells to migrate through 3 µm constrictions. Quantification of the percentage of T cell passage through 3 µm constrictions within
*Figure 7 continued on next page*

*Figure 7 continued*

microchannel devices. (**C**) Example images of WT LifeAct-GFP (Left panels and *Video 3*) and FMNL1 KO/LifeAct-GFP (Right panels and *Video 4*) T cells engaging microchannel constrictions. Top images, LifeAct-GFP (green) and Hoechst (DNA, blue) overlaid on the brightfield channel (gray). Bottom images, pseudocolor rendition of the LifeAct-GFP channel. White dashed arrows indicate cell direction, red arrows point to areas of F-actin accumulation at the back of the cell, black arrows indicate F-actin accumulation at the front of the cell. Time is min:sec, white scale bar = 10 µm. (**D**) FMNL1 promotes actin polymerization at the back of the nucleus during migration under confinement. Top, example of the image masking process to quantify F-actin distribution relative to the front and back of the nucleus. Bottom, quantification of the back-to-front ratio of F-actin distribution during unconfined and confined migration. (**E**) Paired analysis of individual T cells undergoing unconfined and confined migration. Data for cells that increase their F-actin back-to-front ratio under confinement are shown in black, data for cells that decrease the back-to-front ratio are in red. Data in B are the mean ± SEM from 3 independent experiments with a total of 83 control and 68 KO cells analyzed. Data in D are the mean ± SEM and data in E are pooled from 3 independent experiments with a total of 47 control and 35 KO cells analyzed. Statistics in B and E calculated using a two-tailed paired t-test; statistics in D calculated using One-way ANOVA with Sidak's multiple comparisons. n.s. = not significant.

The online version of this article includes the following source data for figure 7:

**Source data 1.** Data points for the graphs in *Figure 7D* are provided as an Excel spreadsheet.

*Dupré et al., 2015*; *Ridley, 2011*). Surprisingly, despite displaying reduced actin polymerizing capacity in response to chemokines, FMNL1-deficient T cells were equivalent to WT cells in their ability to adhere and crawl on to the endothelial monolayer and initiate diapedesis. These observations differ from previous studies of macrophages where FMNL1 was reported to promote the adhesion and migration of macrophages through the formation of membrane protrusions known as podosomes (*Miller et al., 2017*; *Yayoshi-Yamamoto et al., 2000*). While similar membrane structures in T cells have been suggested to play an important role in sensing of chemokine and initiating diapedesis (*Dupré et al., 2015*), our observations of normal crawling and diapedesis initiation in FMNL1-deficient T cells suggest that FMNL1 may not be required for these functions in T cells. The cellular localization of FMNL1 also appears different between macrophages and T cells, with FMNL1 enriched in ventral podosomal structures in macrophages (*Miller et al., 2017*) but enriched behind the nucleus in T cells during TEM as shown herein. These observations argue for a distinct or additional role for FMNL1 in T cells compared to macrophages.

Completion of diapedesis also relies on actin remodeling and force generation as our previous work with the actin polymerizing Ena/VASP proteins and the cytoskeletal motor protein MyoIIA has demonstrated (*Jacobelli et al., 2013*; *Estin et al., 2017*). Similar to Ena/VASP proteins, FMNL1 is dispensable for the early stages of T cell TEM, instead being required for efficient completion of diapedesis. Our data also suggest that FMNL1, like Ena/VASP proteins, has a selective role in the trafficking of activated cells. Despite both Ena/VASP proteins and FMNL1 being propagators of linear actin filaments, they appear to have distinct mechanisms of action. Whereas Ena/VASP proteins mediate diapedesis through regulating the expression and function of α4 integrin (*Estin et al., 2017*), our data suggest that FMNL1 mediates diapedesis by promoting transmigration of the nucleus. Thus, while Ena/VASP deficiency impairs the trafficking of activated T cells to both lymphoid tissue and inflammatory sites, where α4 integrin can be involved (*Issekutz, 1991*), FMNL1 deficiency only impairs trafficking to non-lymphoid tissue sites with more restrictive endothelial barriers. These observations suggest that the nature of the vascular bed through which T cells transmigrate modulates the requirement for FMNL1 and can, in part, explain why FMNL1-deficiency affects effector T cells but not naïve T cells. Endothelial vascular barriers such as in HEVs in lymph nodes are more permissive compared to vascular barriers in peripheral tissues, which form more complex and restrictive tight junctions. This barrier difference likely requires additional force generation by the T cell to push its rigid nucleus through these restrictive barriers, a process that more heavily relies on FMNL1. This finding is supported by our transwell and microchannel migration data, in which the role for FMNL1 appears to be pore size dependent, consistent with the idea that FMNL1 is involved in promoting the transmigration of the rigid nucleus during migration through restrictive barriers.

MyoIIA has also been reported to facilitate nuclear transmigration in T cells, providing contractile force to deform the rigid nuclear structure during both TEM and in migration through restrictive environments (*Jacobelli et al., 2013*; *Friedl et al., 2011*). FMNL1 displays a perinuclear localization during TEM similar to MyoIIA, and we identified a role for FMNL1 in generating perinuclear actin structures during migration under confinement. Analogous to MyoIIA-deficient T cells, FMNL1-deficient T cells also have impaired nuclear transmigration and completion of diapedesis. Thus, FMNL1

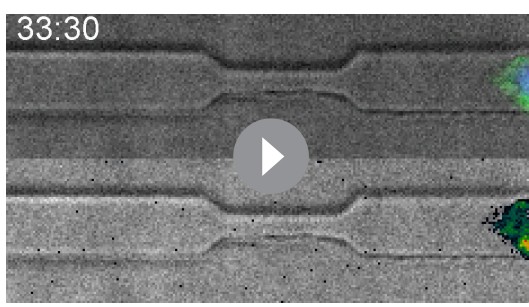

**Video 3.** Example of a control LifeAct-GFP T cell displaying posterior perinuclear actin polymerization while migrating through a microchannel constriction. Control LifeAct-GFP T cells were activated ex-vivo, labeled with Hoechst to visualize their nuclei, and added to microchannel devices. T cells crawling in the microchannels were imaged at 37°C for 2–6 hr. Brightfield and fluorescent images were acquired every 30 s. Top images, LifeAct-GFP (green) and Hoechst (DNA, blue) overlaid on the brightfield channel (gray). Bottom images, pseudocolor rendition of the LifeAct-GFP channel. This representative control LifeAct-GFP T cell shows posterior perinuclear actin polymerization while migrating through a microchannel constriction. Time is displayed as min:s.

https://elifesciences.org/articles/58046#video3

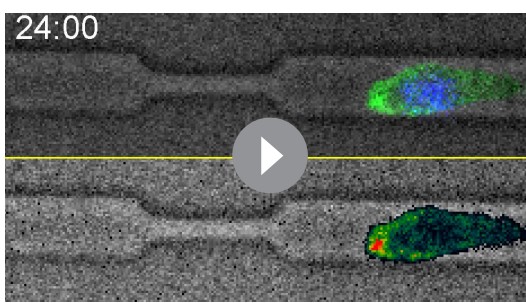

**Video 4.** Example of a FMNL1 KO/LifeAct-GFP T cell failing polymerize actin at the back of the cell and unable to migrate through a microchannel constriction. FMNL1 KO/LifeAct-GFP T cells were activated ex-vivo, labeled with Hoechst to visualize their nuclei, and added to microchannel devices. T cells crawling in the microchannels were imaged at 37°C for 2–6 hr. Brightfield and fluorescent images were acquired every 30 s. Top images, LifeAct-GFP (green) and Hoechst (DNA, blue) overlaid on the brightfield channel (gray). Bottom images, pseudocolor rendition of the LifeAct-GFP channel. This representative FMNL1 KO/LifeAct-GFP T cell lacks posterior perinuclear actin polymerization as its nucleus engages the microchannel constriction and is unable to migrate through the constriction. Time is displayed as min:s.

https://elifesciences.org/articles/58046#video4

may be acting either in parallel or in cooperation with MyoIIA to facilitate nuclear transmigration. One possibility is that FMNL1 is generating an actin network that is then crosslinked by MyoIIA to generate contractile force. Alternatively, FMNL1 may be generating different actin structures to provide propulsion force for nuclear transmigration. In addition to its actin polymerizing function, FMNL1 has also been implicated in actin bundling and severing activities (*Harris et al., 2004*; *Harris et al., 2006*) and could remodel actin networks in various ways to promote TEM.

Previous investigations of mDia1, the other highly-expressed formin in T cells, observed defects in T cell development and trafficking to secondary lymphoid organs (*Sakata et al., 2007*; *Eisenmann et al., 2007*). In contrast, in FMNL1-deficient mice we observed normal populations of both developmental and mature T cells and identified a selective requirement for FMNL1 extravasation through restrictive endothelial barriers and trafficking to inflamed tissues. Furthermore, while prior reports indicate that loss of mDia1 impairs chemotaxis even through non-restrictive barriers (*Thompson et al., 2018*; *Sakata et al., 2007*), we observed that FMNL1 was selectively required for chemotaxis through narrow pores. Together these findings suggest a distinct and specific role for FMNL1 in mediating nuclear transmigration through restrictive endothelial barriers compared to a more general role for mDia1 in T cell motility.

The use of FMNL1 by T cells to promote transmigration of the nucleus may represent a unique mechanism compared to those employed by other immune cell types. For example, dendritic cells have been suggested to use actin polymerization by Arp2/3, instead of by formins, to facilitate nuclear deformation when migrating through restrictive environments (*Thiam et al., 2016*). Nuclear flexibility may also dictate the requirement for actin mediated nuclear manipulation. It has been reported that neutrophils, which have multi-lobed deformable nuclei, or cells with induced nuclear flexibility do not display perinuclear actin accumulation when encountering restrictive spaces, while cells with rigid nuclei do (*Thiam et al., 2016*). Consistent with this literature, we observed that FMNL1 was dispensable for neutrophil migration through narrow pores, while in activated T cells, varying the level of FMNL1 expression strongly correlated with narrow pore transmigration. After activation, the nuclear volume of T cells increases but still maintains a rigid ovoid structure in contrast to the

flexible segmented nuclear structure of neutrophils (*Gupta et al., 2012*; *Friedl et al., 2011*). However, similar to neutrophils, effector T cells must rapidly migrate into tissues to execute their functions (*Masopust and Schenkel, 2013*). Thus, we propose that FMNL1-driven perinuclear actin polymerization in activated T cells compensates for their increased nuclear size and rigidity by facilitating nuclear migration through constricted environments such as the restrictive endothelial barriers of non-lymphoid tissues.

Our observations of T cell induced EAE and type 1 diabetes models suggest that reduced trafficking by FMNL1-deficient T cells to sites of inflammation plays a role in impairing the induction of autoimmune disease. While we observed a partial but significant inhibition of trafficking, we observed complete disease protection in a subset of mice in both disease phenotypes. The differences between these findings may be due to the fact that in the activated T cell transfer disease models, no prior immune infiltration has occurred, and thus, the vascular endothelial barriers are much more impermissive compared to the trafficking models where inflammation and immune cell recruitment is ongoing. Furthermore, there may be a threshold effect whereby trafficking is reduced such that the number of FMNL1-deficient T cells capable of entering the tissue are not sufficient to induce disease. While, in this report, we have examined the function of FMNL1 in T cell TEM, it is also possible that, in these disease contexts, FMNL1 may have additional roles within the disease site or might impact migration within restrictive tissue parenchyma. It has been previously reported that knockdown of FMNL1 slightly impaired human CD8 T cell killing in vitro (*Gomez et al., 2007*), thus FMNL1 deficiency may also play a role in OT-I T cell effector function in the RIP-mOVA model. Future study of FMNL1 function in T cells within tissues may identify additional mechanisms by which FMNL1 regulates T cell behavior.

Inhibition of T cell trafficking has shown therapeutic promise in the treatment of autoimmune and inflammatory disease (*Luster et al., 2005*; *Engelhardt and Kappos, 2008*). Our observations that FMNL1 deficiency in T cells impairs their ability to traffic to autoimmune inflammatory sites and induce autoimmune disease in both EAE and type 1 diabetes suggest that FMNL1 may be a potential therapeutic target for autoimmune disease treatment. As immune cell development and homeostatic trafficking appear unaltered by global FMNL1 deficiency, targeting of FMNL1 may offer the possibility of selectively impairing activated T cell trafficking to specific tissue sites while leaving systemic immune surveillance intact. A recent report has suggested that selectively inhibiting activated T cell trafficking may be particularly beneficial in the context of graft-versus-host disease (*Huang et al., 2015*). While previously studied cytoskeletal effectors of T cell migration, such as Arp2/3, mDia1, MyoIIA and Ena/Vasp proteins are expressed in most tissues and cell types, FMNL1 expression is largely restricted to hematopoietic cells (*Yayoshi-Yamamoto et al., 2000*; *Gardberg et al., 2014*; *Yue et al., 2014*). Additionally, FMNL1 expression has been reported to be transcriptionally upregulated by T cells that migrate to sites of autoimmune inflammation (*Odoardi et al., 2012*; *Degroote et al., 2017*). This expression profile may offer the possibility of a more targeted approach to therapeutically alter cytoskeletal function in immune cells.

In conclusion, our observations have uncovered a new mechanism that promotes lymphocyte nucleus passage through constrictive environments. This process of FMNL1-dependent nucleus transmigration is key for the extravasation of effector T cells through the restrictive vascular endothelial barriers present in non-lymphoid tissues. Furthermore, FMNL1 deficiency in T cells impairs their ability to induce autoimmune disease, suggesting that targeting of FMNL1 may be therapeutically beneficial.

## Materials and methods

**Key resources table**

| Reagent type (species) or resource | Designation | Source or reference | Identifiers | Additional information |
|---|---|---|---|---|
| Genetic Reagent (*M. musculus*) | FMNL1 KO | This Paper | | FMNL1 deficient mouse on C57BL/6 genetic background |

*Continued on next page*

*Continued*

| Reagent type (species) or resource | Designation | Source or reference | Identifiers | Additional information |
|---|---|---|---|---|
| Genetic Reagent (*M. musculus*) | LifeAct-GFP | *Riedl et al., 2008* | RRID:IMSR_EM:12427 | Mice expressing LifeAct-GFP construct on C57BL/6 genetic background |
| Antibody | anti-FMNL1 (mouse monoclonal) | Santa Cruz Biotechnology | Cat#: sc-390466 RRID:AB_2847962 Clone A-4 | IF (1:100) |
| Recombinant DNA Reagent | FMNL1-IRES-GFP MMLV (plasmid) | VectorBuilder | | pMMLV vector that expresses FMNL1 and GFP (under an IRES) |
| Recombinant DNA Reagent | Control GFP MMLV and Control RFP MMLV (plasmids) | VectorBuilder | | Control pMMLV vectors that express GFP or RFP |
| Other | Microchannel fluidic devices | 4DCell | Cat#s: MC011 and MC019 | Microchannel devices with variable channel widths and constrictions |
| Cell line (*M. musculus*) | bEnd.3 | ATCC (*Montesano et al., 1990*) | Cat#: CRL-2299 RRID:CVCL_0170 | Mouse endothelial cell line used in the TEM assays |

## Generation of FMNL1 KO mice

To obtain FMNL1 KO mice, we targeted the *Fmnl1* gene via homologous recombination by inserting, after exon 3 of *Fmnl1*, a strong splice acceptor and a LacZ tag followed by a Neomycin selection cassette into C57BL/6 embryonic stem (ES) cells. The targeting construct was purchased from EUCOMM. This construct also inserts loxP sites flanking exons 4 to 6 as well as FRT sites flanking the LacZ tag and Neomycin cassette. This can enable future conversion of the targeted allele into a conditional KO after crossing with Flp recombinase mice to eliminate the LacZ-Neomycin cassette and then crossing to mice expressing Cre recombinase to excise exons 4–6. To target the *Fmnl1* allele, C57BL/6N (JM8.F6) ES cells were electroporated with the targeting construct and selected using G418. Individual ES cell colonies were isolated, and putative positive homologous recombinants were identified using loss of allele analysis followed by long range PCR through both the 5' and 3' ends of the targeting construct into the *Fmnl1* locus. The homologous recombinant ES cell clones were chromosome counted, and clones with the best euploidy were microinjected into C57BL/6 albino blastocysts. High percentage male chimeras generated from these injections were bred to C57BL/6 albino females. Offspring with black coat color were genotyped for the desired *Fmnl1* targeting by PCR. Mice containing the *Fmnl1* targeted allele were selected as breeders to propagate the FMNL1 KO transgenic line. Targeting of the *Fmnl1* gene was further confirmed by western blot comparing lysates from T cells derived from control and homozygous FMNL1 KO mice, which show a complete loss of FMNL1 protein.

## Mice

For characterization experiments and experiments with polyclonal T cells, FMNL1 KO mice were paired with age and sex matched WT C57BL/6 mice bred in house. FMNL1 KO mice were crossed with OT-I (C57BL/6-Tg(TcraTcrb)1100Mjb/J, RRID:IMSR_JAX:003831), OT-II (B6.Cg-Tg(TcraTcrb) 425Cbn/J, RRID:IMSR_JAX:004194), or 2D2 (C57BL/6-Tg(Tcra2D2,Tcrb2D2)1Kuch/J, RRID:IMSR_JAX:006912) TCR transgenic mice bred in house to create respective FMNL1 KO TCR transgenic mice. FMNL1 KO mice were also bred with LifeAct-GFP mice (*Riedl et al., 2008*) (RRID:IMSR_EM: 12427) a generous gift from Dr. Roland Wedlich-Söldner (University of Münster). Age and sex matched WT TCR transgenic mice bred in house were used as controls. CD45.1/.1 (B6.SJL-*Ptprc*[a] *Pepc*[b]/BoyJ, RRID:IMSR_JAX:002014) mice were purchased from Charles River and bred in house. RIP-mOva (C57BL/6-Tg(Ins2-TFRC/OVA)296Wehi/WehiJ, RRID:IMSR_JAX:005431) mice were purchased from Jackson Labs. All mice used in experiments were 8–16 weeks old. All experiments

involving mice were approved by the Institutional Animal Care and Use Committees of National Jewish Health (Protocol #AS2811-01-23) and the University of Colorado School of Medicine (Protocol #000937). All efforts were made to minimize mouse suffering.

## Cell lines

The mouse endothelial cell line bEnd.3 (*Montesano et al., 1990*) (from ATCC, RRID:CVCL_0170) was used in the TEM assays. The endothelial identity of this cell line was confirmed by checking the expression of von Willebrand factor and uptake of fluorescently labeled low density lipoprotein (LDL). We further confirmed the expression of endothelial adhesion molecules (e.g. ICAM-1 and VCAM-1) on the surface of bEnd.3 cells and lack of mycoplasma.

## Media

T cells were cultured in R10 media: RPMI 1640 (Corning) supplemented with L-glutamine (292 μg/mL), penicillin (100 U/mL), streptomycin (100 μg/mL), and β-mercaptoethanol (50 μM) (all purchased from Gibco) and 10% Fetal Bovine Serum (FBS, Corning lot # 35010150). The bEnd.3 endothelial cell line was cultured in D10 media: DMEM (Corning) supplemented with 10 mM HEPES (Corning), L-glutamine, penicillin, streptomycin, and β-mercaptoethanol and 10% Fetal Bovine Serum (as above). For microscopy experiments cells were resuspended in imaging media: RPMI 1640 without phenol red (Gibco) supplemented with 2% bovine serum albumin (BSA, Millipore-Sigma) and 10 mM HEPES (Corning). Imaging media was also used for actin polymerization and transwell assays.

## Antibodies for flow cytometry

The following monoclonal antibodies were used for flow cytometry staining: CD4 (GK1.5), CD8a (53–6.7), CD3 (145–2 C11), CD19 (6D5), NK1.1 (PK136), CD11b (M1/70), Ly6C (HK1.4), Ly6G(1A8), Siglec F(E50-2440), CD11c (N418), MHCII IA/IE (M5.114.15.2), F4/80 (BM8), CD64 (X54-5/7.1), CD44 (IM7,), CD62L (Mel14), CD49d (R1.2), CD11a (M17/4), CXCR3 (CXCR3-173), CXCR4 (L276F12), CD45.1 (A20), CD45.2 (104), CD45 (30-F11). Clone identifier is indicated in parentheses. These antibodies were purchased from Biolegend, eBioscience, or BD Bioscience. Prior to staining for all experiments involving flow cytometry, cells were blocked with anti-CD16/CD32 antibodies (2.4G2, BioXcell).

## Western blotting

FMNL1 was detected using either a goat polyclonal antibody (Santa Cruz) or rabbit polyclonal antibody (Abcam). As a loading control, α-tubulin was detected using a mouse monoclonal antibody (B-1-5-2, Millipore Sigma). Antibody staining was visualized using the Odyssey near-infrared imaging system with IRDye-680 or-800 secondary antibodies (Li-cor Biosciences). Band intensities were quantified with densitometry using Odyssey 2.1 software (Li-cor Biosciences).

## Flow cytometry characterization of FMNL1 KO mice

Thymus, blood, spleen, and inguinal lymph nodes were collected from FMNL1 KO mice or age and sex matched WT C57BL/6 mice. Blood was lysed in 175 mM ammonium chloride (Millipore Sigma) for 30 min on ice. Spleen and LN were mechanically dissociated and then digested with Collagenase D (0.4 Wunsch U/mL) and DNase (250 μg/mL) (both from Roche) for 30 min at 37°C. After preparations of single cell suspensions, cells were stained with fluorescently-labeled antibodies. For quantification of cell numbers, a known number of CountBright Absolute Counting Beads (ThermoFisher Scientific) were added to each sample. Samples were then analyzed by flow cytometry on a BD LSR Fortessa. Thymic populations were identified as follows: CD4 single positive (SP): CD4$^+$, CD8$^-$; CD8 SP: CD4$^-$, CD8$^+$; double positive (DP): CD4$^+$, CD8$^+$; double negative (DN) CD4$^-$, CD8$^-$. Lymphocyte populations were identified as follows: CD4 T cells: CD3$^+$, CD4$^+$; CD8 T cells: CD3$^+$, CD8$^+$; B Cells: CD3$^-$, CD19$^+$; natural killer (NK) cells: CD3$^-$, NK1.1$^+$. Myeloid populations were identified as follows: monocytes: CD11b$^+$, Ly6C$^{high}$; eosinophils: CD11b$^+$, Siglec F$^+$; neutrophils CD11b$^+$, Ly6G$^+$; dendritic cells (DCs): CD11c$^+$, MHCII$^{high}$; macrophages: CD11b$^+$, Ly6C$^{int}$, F4/80$^+$, CD64$^+$.

### T cell isolation

Naive CD4, total CD4, or total CD8 T cells were isolated from dissociated and pooled spleen, axillary, brachial, inguinal and mesenteric lymph nodes using magnetic negative selection kits (Stemcell Technologies) as described previously (*Estin et al., 2017*). Naive CD4 T cells were used for naive T cell trafficking experiments. CD8 T cells were used for OT-I experiments, and islet trafficking experiments. CD4 T cells were used for all other trafficking experiments, microscopy experiments, and in vitro experiments.

### Polyclonal T cell activation and culture

Polyclonal T cells were activated ex vivo as described previously (*Estin et al., 2017*). Briefly, isolated T cells were activated with plate-bound anti-CD3 (2C11, 2 µg/well of 24 well plate) and soluble anti-CD28 (PV-1) (2 µg/mL) antibodies (both from BioXCell) in the presence of irradiated CD45.1/.1 feeder splenocytes for two days in R10. Cells were then removed from the plate and cultured with 10 U/mL recombinant human interleukin 2 (rIL2, AIDS Research and Reference Reagent Program, Division of AIDS, National Institute of Allergy and Infectious Diseases, National Institutes of Health, from M. Gately, Hoffmann-La Roche) for 3 days with addition of fresh R10 media and rIL2 on day 4 post-activation. By day 5, all CD45.1/.1 irradiated splenocytes have died. Prior to use dead cell and debris were removed from the culture by Histopacque-1119 (Millipore Sigma) density gradient.

### Fluorescent Dye-Labeling of T cells

For lymphoid organ and islet trafficking experiments, WT or FMNL1 KO T cells were differentially dye-labeled with either Cell Proliferation Dye eFluor670 (ThermoFisher Scientific) or Violet Proliferation Dye 450 (VPD) (BD Biosciences) and then mixed at a 1:1 ratio prior to transfer as described previously (*Lindsay et al., 2015*). For CNS trafficking experiments WT or FMNL1 KO T cells were differentially dye-labeled with VPD or CFSE (ThermoFisher Scientific) and then mixed at a 1:1 ratio prior to transfer as described previously (*Estin et al., 2017*). For microscopy experiments WT and FMNL1 T cells were differentially dye-labeled with either CFSE or CellTrace Yellow (CTY) (ThermoFisher Scientific) and mixed at a 1:1 ratio. For all experiments involving dye-labeled T cells, between experimental repeats, dyes were swapped between WT and FMNL1 KO T cells to control for potential effects of the dyes.

### Lymphoid organ T cell trafficking

WT and FMNL1 KO T cells were differentially dye-labeled, mixed at a 1:1 ratio ($2.5 \times 10^6$ cells each) ratio and transferred intravenously (i.v.) into CD45.1/.1 recipient mice. 24 hr post-transfer mice were euthanized with $CO_2$. Blood was collected by cardiocentesis and red blood cells were lysed for 30 min on ice with 175 mm ammonium chloride. Spleen was collected and mechanically dissociated and red blood cells were lysed for 5 min at room temperature in 175 mm ammonium chloride. Inguinal lymph nodes were collected and mechanically dissociated. After preparation of single-cell suspensions, isolated cells were stained with anti-CD45.2 antibodies. For quantification of cell numbers, a known number of CountBright Absolute Counting Beads (ThermoFisher Scientific) were added to each sample. Samples were then analyzed on a CyAn ADP flow cytometer (Beckman Coulter). Transferred T cells were identified as CD45.2$^+$ and either VPD$^+$ or efluor670$^+$.

### Islet trafficking

OT-I T cells ($10^7$) were isolated and transferred i.v. into RIP-mOVA recipient mice 7 days prior to harvest to induce immune infiltration of the islets. Ex vivo activated, polyclonal, WT and FMNL1 KO CD8$^+$ T cells were differentially dye-labeled, mixed at a 1:1 ratio ($10^7$ cells each) and transferred i.v. into these RIP-mOVA mice 24 hr prior to harvest. Mice were anesthetized with intraperitoneal (i.p.) ketamine (50 mg/g, Vedco) and xylazine (5 mg/g, JHP) prior to euthanasia by cardiocentesis and cervical dislocation. Blood was collected and processed as above. Pancreatic islets were harvested as described previously (*Friedman et al., 2014*; *Lindsay et al., 2015*). Briefly, the pancreas was inflated via the common bile duct with ~3 mL of 0.8 mg/mL Collagenase P (Roche) and 10 µg/mL Dnase I (Roche) in HBSS (Corning). Following inflation, the pancreas was removed and incubated at 37°C for 10–16 min, and the islets were isolated by density centrifugation. Intact islets were then handpicked under a dissecting microscope, and subsequently digested with Collagenase D (0.4 Wunsch U/mL,

Cell Biology | Immunology and Inflammation

Roche) for 30 min, followed by 30 min in Cell Dissociation Buffer (Millipore Sigma) to prepare a single cell suspension. Isolated cells were stained with anti-CD45 (to exclude any non-hematopoietic cells isolated from the islets), and anti-CD8 antibodies. For quantification of cell numbers, a known number of CountBright Absolute Counting Beads (ThermoFisher Scientific) were added to each sample. Samples were then analyzed on a BD LSR Fortessa. Transferred T cells were identified as CD45$^+$, CD8$^+$, and either efluor670$^+$ or VPD$^+$.

## CNS trafficking
First, EAE was induced in recipient mice using induction kits (Hooke Laboratories) according to the manufacturers protocol. Briefly, WT female CD45.1/.1 mice were immunized with MOG35–55 peptide emulsified in complete Freund's adjuvant injected subcutaneously, followed by intraperitoneal injection of pertussis toxin on the day of induction and the following day. EAE onset was within 10–15 d post-immunization. Mice were monitored and scored daily for development of EAE based on the following 0–5 scoring criteria: 0, no disease; 1 limp tail; 2, weakness or partial paralysis of hind limbs; 3, full paralysis of hind limbs; 4, complete hind limb paralysis and partial front limb paralysis; 5, complete paralysis of front and hind limbs or moribund state. Mice with a score ≥4 were euthanized immediately. Experiments to quantify trafficking to the brain and spinal cord were performed as described previously (*Estin et al., 2017*). Briefly, ex vivo activated, polyclonal WT and FMNL1 KO CD4$^+$ T cells were differentially dye-labeled, mixed at a 1:1 ratio (10$^7$ cells each) and transferred i.v. into CD45.1/.1 mice with an EAE score of 2.0–3.0. 24 hr later mice were euthanized for tissue collection. To distinguish transferred cells in the vasculature from those fully extravasated into the parenchyma of tissues, 4 min prior to euthanasia, mice were injected via tail vein with 3 μg of anti-CD4-allophycocyanin (APC) as described previously (*Anderson et al., 2014*). After euthanasia, blood was collected by cardiocentesis and the vasculature of the mouse was perfused via saline through the heart. Blood was processed as above. Brain and spinal cord were mechanically dissociated and total leukocytes were then isolated with a 70%/30% Percoll gradient (Millipore Sigma). After preparation of single-cell suspensions, isolated cells were stained with anti-CD45.2, and anti-CD45.1 antibodies. For quantification of cell numbers, a known number of CountBright Absolute Counting Beads (ThermoFisher Scientific) were added to each sample. Samples were then analyzed on a BD LSR Fortessa. Transferred T cells were identified as CD45.2$^+$, CD45.1$^-$, and either CFSE$^+$ or VPD$^+$. In the brain and spinal cord samples, extravasated cells were identified as intravascular CD4$^-$.

## T cell transfer diabetes model
WT and FMNL1 KO CD8$^+$ OT-I T cells were activated ex vivo with OVA peptide 257–264 (Pi Proteomics) for 2 days in the presence of irradiated CD45.1/.1 splenocytes and then cultured with IL-2 as above for 4 days. WT and FMNL1 KO CD4$^+$ OT-II T cells were activated with OVA peptide 323–339 (GenScript) for 2 days in the presence irradiated CD45.1/.1 splenocytes and then cultured with IL-2 as above for 4 days. On day 6 post-antigen stimulus, dead cells were removed by Histopacque-1119 (Millipore Sigma) density gradient. OT-I (5 × 10$^6$) and OT-II (2.5 × 10$^6$) T cells of the same FMNL1 genotype (WT with WT, KO with KO) were then combined and transferred i.v. into RIP-mOVA recipient mice. The blood glucose of the recipient mice was then monitored daily from day 4–28 post-transfer. Mice with blood glucose levels of greater than 350 mg/dL on two consecutive days were considered to be diabetic and were euthanized.

## T cell transfer EAE model
WT and FMNL1 KO CD4$^+$ 2D2 T cells were activated with MOG peptide 35–55 (CHI Scientific) for two days in the presence of irradiated CD45.1/.1 splenocytes and then cultured with IL-2 as above for 5 days. On day 7 post-antigen stimulus, dead cells were removed by Histopacque-1119 (Millipore Sigma) density gradient. Cells were then restimulated with plate-bound anti-CD3 (2C11, 1 μg/well of 24 well plate) and plate-bound anti-CD28 (PV-1, 1 μg/well of 24 well plate) antibodies (both from BioXCell) for two days. Cells were then removed from the plate and rested for an additional day in R10. 5 × 10$^6$ of these activated WT or FMNL1 KO 2D2 T cells were then transferred i.v. into sublethally irradiated (300 rads) CD45.1/.1 recipient mice. From days 7–28 post-transfer, mice were monitored and scored for development of EAE as described above. The researchers performing the disease scoring were blinded as to the experimental group of the mice.

## Microscopy

All microscopy experiments employed a 3i (Intelligent Imaging Innovations) Marianas spinning-disk confocal microscope system equipped with a Zeiss inverted stand and a Yokogawa spinning disk unit. The microscope is housed in an environmental control chamber and all imaging of live-cells was performed at 37°C.

## T cell TEM under flow

TEM experiments were performed as described previously (*Estin et al., 2017*; *Thompson et al., 2018*; *Wigton et al., 2016*). Briefly, a 1:1 mixture of ex vivo activated, differentially fluorescently dye-labeled WT and FMNL1 KO T cells was resuspended in imaging media at $2 \times 10^6$ cells/mL. T cells were perfused at 0.2 dyne/cm$^2$ shear-flow into a flow chamber (μ-slide VI, IBIDI) coated with a monolayer of bEnd.3 brain-derived endothelial cells, activated with 40 ng/mL TNF-α (ThermoFisher Scientific) 24 hr before imaging and treated with 10 ng/mL CXCL10 (Peprotech) 30 min before imaging. After 5 min of accumulation, the shear-flow was raised to 2 dyne/cm$^2$. Phase contrast and fluorescence images were acquired every 20 s for 30 min using a spinning-disk confocal microscope with a 20x phase objective (Zeiss). Time-lapse images were then analyzed using SlideBook 6 (3i) and the stages of TEM manually scored as described previously (*Estin et al., 2017*; *Thompson et al., 2018*; *Wigton et al., 2016*). Briefly, T cells that lose a portion of the phase halo localized to a small protrusion of the cell, detected by fluorescence, are considered to have attempted diapedesis, with completion being scored as complete loss of the phase halo. Diapedesis data were filtered to exclude all cells that were not present in the field of view for at least 13 min (twice the mean time from the start of imaging until the completion of diapedesis). This filtering is applied to prevent biased scoring of cells that were not present in the imaging field of view for very long as being failed attempts at diapedesis.

## FMNL1 localization during TEM

Ex vivo activated WT T cells dye-labeled with CTY were perfused into flow chambers containing bEnd.3 monolayers for 5 min as above. Cells were allowed to migrate under 2 dyne/cm$^2$ shear flow for 5 min and then fixed with 4% (w/v) paraformaldehyde (Electron Microscopy Sciences) in PBS for 10 min. Cells were then permeabilized and blocked for 1 hr at room temperature with the following saponin buffer: 0.5% (w/v) saponin (Millipore-Sigma) 2% (v/v) FBS (Corning) 2% (v/v) normal donkey serum (Jackson ImmunoResearch Laboratories) with 0.05% (w/v) sodium azide (Millipore Sigma) in PBS (Millipore-Sigma). Cells were stained overnight at 4°C with a mouse monoclonal antibody against FMNL1 (A-4, Santa Cruz) diluted 1:100 in the saponin solution. To visualize the FMNL1 antibody, cells were subsequently stained with a Dylight-649 conjugated Donkey anti-mouse secondary antibody (Jackson ImmunoResearch) diluted 1:100 in saponin buffer for 1 hr at room temperature. To visualize the nucleus, cells were additionally stained with 250 ng/mL DAPI (ThermoFisher Scientific) during the secondary antibody step. Multi-plane images were then acquired across a 10 μm range using a spinning-disk confocal microscope with a 40x Water/Oil objective (Zeiss). Maximum Z-projections and side-view reconstruction images were compiled using SlideBook 6 (Intelligent Imaging Innovations). Linescan quantification of fluorescence intensities of FMNL1 and nucleus staining along the axis of T cells undergoing transendothelial migration was performed using ImageJ (NIH). After background fluorescence subtraction, the linescan intensities were then used to categorize the level of FMNL1 enrichment adjacent to the back of nucleus as follows: perinuclear enrichment: $\geq$1.33x fold peak of FMNL1 fluorescence intensity behind the nucleus compared to rest of the cell; no enrichment: lack of a main peak of FMNL1 fluorescence behind the nucleus; partial enrichment: presence of a FMNL1 peak behind the nucleus with additional peaks of similar intensities in other locations of the cell.

## Nucleus localization during TEM

Ex vivo activated, differentially dye-labeled WT and FMNL1 KO T cells were perfused into flow chambers with bEnd.3 monolayers, allowed to migrate, and then fixed as above. Cells were stained with DAPI as above and then imaged using a spinning-disk confocal microscope (3i) with a 40x phase objective (Zeiss). Similar to the live cell analysis above, phase contrast imaging was analyzed with SlideBook 6 and manually scored to determine the position of the nucleus relative to the plane of the

endothelium as follows: above: nucleus completely surrounded by the phase halo; in process: nucleus partially surrounded by the phase halo; below: nuclei with complete absence of the phase halo.

## T cell transwell migration

Wells of a 24-well plate were prepared containing either 5 µm or 3 µm transwell inserts (Corning) and imaging media with either 100 ng/mL CXCL10 or 1 ug/mL CXCL12 (both from Peprotech) in the bottom chamber. $1 \times 10^6$ WT or FMNL1 KO T cells were added to the top chambers and allowed to migrate for 1 hr at 37°C into the lower well. $2 \times 10^5$ cells (20% of input cells added to transwells) were placed directly into bottom wells with no transwell as a standard to calculate the percentage of migrated cells. Each condition was set up in duplicate. Migrated T cells were collected from the bottom wells and 25 µL of CountBright Absolute Counting Beads (ThermoFisher Scientific) were added to each sample to enable quantification of migrated cells. Each sample was quantified for a fixed period (30 s) using a flow cytometer (CyAn ADP Beckman Coulter). The number of cells counted during this time was normalized to the number of beads counted to adjust for any variations in flow rate during the run.

## Neutrophil transwell migration

Wells of a 24 well plate were prepared with 3 µm transwell inserts as above and with 100 ng/mL CXCL1 (Peprotech) in the bottom chamber. Neutrophils were isolated from the bone marrow of WT or FMNL1 KO mice using magnetic negative selection kits (StemCell Technologies) and then $1 \times 10^6$ cells were added to the top chambers of the transwells. Cells were allowed to migrate for 1 hr at 37°C and then quantified as above.

## Actin polymerization assay

Quantification of actin polymerization in response to chemokine was performed as described previously (*Estin et al., 2017*). Briefly, ex vivo activated WT or FMNL1 KO T cells were stimulated with either no chemokine, 1 µg/mL CXCL12 or 100 ng/mL CXCL10 (Peprotech) for 5, 15, or 60 s at 37°C in imaging media. All conditions were set up in duplicate. The reaction was stopped using 4% (w/v) paraformaldehyde (Electron Microscopy Sciences) in PBS and the cells were fixed for 10 min. T cells were then permeabilized with saponin buffer (as in microscopy experiments) for 30 min at room temperature. Cells were stained with a 1:50 dilution of Phalloidin Alexa Fluor 647 (ThermoFisher Scientific) and fluorescence was quantified using a CyAn ADP flow cytometer (Beckman Coulter). Geometric mean fluorescence intensities (gMFI) were normalized to the values of the unstimulated samples.

## FMNL1 Re-expression

WT or FMNL1 KO T cells were activated with anti-CD3 and anti-CD28 antibodies as above and then on day two post-activation transduced with Maloney murine leukemia retrovirus (MMLV) constructs either expressing FMNL1 and GFP (under an IRES) or fluorescent protein alone. To maximize expression, cells were transduced a second time with the same virus on day 3 post-activation. On day four post-activation, GFP$^+$ T cells were sorted using an ICyte Synergy (Sony). After 24 hr of culture, cells were used in transwell assays or for western blot as described above.

## Microchannel migration imaging

PDMS (polydimethylsiloxane) microchannel fluidic devices with constrictions were purchased from 4DCell (Montreuil, France). Microchannels with 8 µm or 12 µm channel widths and 3 µm constrictions were used. Five to six days after activation, $1-2 \times 10^5$ ex vivo activated control and FMNL1 KO T cells were fluorescently labeled and added to the central entry ports of the microchannel devices and CXCL10 (100 ng/mL) was added to the side ports. T cells were allowed to spontaneously enter the channels at 37°C for 1 hr. T cells crawling in the microchannels were then imaged on a spinning disk confocal at 37°C for 2–6 hr at intervals of 30 or 60 s. In a separate set of experiments to determine F-actin distribution during migration within microchannels with constrictions, control LifeAct-GFP or FMNL1 KO/LifeAct-GFP ex vivo activated T cells were labeled with Hoechst, to visualize their nuclei, and then added to microchannel devices as above. Image acquisition and analysis were done

using Slidebook 6 software to quantify passage through constrictions and actin distribution relative to the T cell nucleus. T cell passage through constrictions and F-actin distribution scoring was performed in a blinded manner. F-actin distribution analysis was performed by generating a mask based on the fluorescence intensity of the LifeAct-GFP channel using a 2-fold over background threshold. The mask was then separated into front and back parts relative to the middle of the nucleus and the direction of migration. The mean fluorescence intensity of LifeAct-GFP in each mask was then determined and a back-to-front intensity ratio calculated. For each analyzed cell, the back-to-front F-actin ratio was determined during unconfined migration and at the initial timepoint at which the cell's nucleus started engaging the 3 µm constriction.

## Statistical analysis

GraphPad Prism 7.0 software was used to create graphs and to perform all statistical analyses. Specific statistical tests and n for each experiment are indicated in the corresponding figure legend. In experiments looking directly at WT or FMNL1 KO mice, such as the characterization experiments, the individual mice were considered to be the independent experimental unit (n). In experiments where WT or KO T cells were transferred into multiple recipient mice, the source mouse for transferred cells was considered the independent experimental unit and statistics were performed on the mean values from mice receiving the same source of cells. Similarly, for microscopy experiments, the source mouse was considered to be the independent experimental unit, and statistics were performed on the mean values from cells from the same source, except for *Figure 7D and E* in which statistics were performed on individual cell values.

## Acknowledgements

We thank B Basta, O Castro-Villasano, K Dew, M Gebert, K Morgan, J Olivas, J Phares, E Rodriguez, D Tracy, and S Yannacone for technical help with mouse genotyping and colony maintenance; J Loomis and S Sobus for expert technical assistance with cell sorting and flow cytometer maintenance; J Loomis for microscope maintenance; Drs. K Haist, R Kedl, RL Reinhardt, L Sussel, and R Torres for reading of the manuscript and feedback.

## Additional information

### Funding

| Funder | Grant reference number | Author |
| --- | --- | --- |
| National Institute of Allergy and Infectious Diseases | R56AI105111 | Jordan Jacobelli |
| National Institute of Allergy and Infectious Diseases | R01AI125553 | Jordan Jacobelli |
| National Institute of Allergy and Infectious Diseases | R21AI119932 | Rachel S Friedman |
| Juvenile Diabetes Research Foundation, International | 5-2013-200 | Rachel S Friedman Jordan Jacobelli |
| National Institute of Allergy and Infectious Diseases | T32AI007405 | Scott B Thompson Monique M Waldman Miriam L Estin |
| National Multiple Sclerosis Society | PP1775 | Jordan Jacobelli |

The funders had no role in study design, data collection and interpretation, or the decision to submit the work for publication. The content of this work is solely the responsibility of the authors and does not necessarily represent the official views of the NIH or other funding agencies.

## Author contributions
Scott B Thompson, Conceptualization, Formal analysis, Investigation, Methodology, Writing - original draft, Writing - review and editing; Adam M Sandor, Victor Lui, Jeffrey W Chung, Monique M Waldman, Robert A Long, Investigation; Miriam L Estin, Investigation, Methodology; Jennifer L Matsuda, Resources, Methodology; Rachel S Friedman, Conceptualization, Methodology; Jordan Jacobelli, Conceptualization, Formal analysis, Supervision, Funding acquisition, Investigation, Methodology, Writing - original draft, Project administration, Writing - review and editing

## Author ORCIDs
Jordan Jacobelli (iD) https://orcid.org/0000-0001-6612-6704

## Ethics
Animal experimentation: All experiments involving mice were approved by the Institutional Animal Care and Use Committees of National Jewish Health (Protocol #AS2811-01-23) and the University of Colorado School of Medicine (Protocol #000937). All efforts were made to minimize mouse suffering.

## Decision letter and Author response
Decision letter https://doi.org/10.7554/eLife.58046.sa1
Author response https://doi.org/10.7554/eLife.58046.sa2

# Additional files

## Supplementary files
• Transparent reporting form

## Data availability
The data generated and analysed in this study are included in the manuscript and/or supporting files.

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
