## [Decision Letter]

**Acceptance summary:**

There was unanimous agreement amongst reviewers and editors that you and your colleagues makes important contributions to unexpected differences in cytoskeletal machinery for entry of T cells into sites of inflammation. The work may, in the future, contribute to treatment of autoimmune and inflammatory diseases.

**Decision letter after peer review:**

Thank you for submitting your article "Formin-like 1 mediates effector T cell trafficking to inflammatory sites to enable T cell-mediated autoimmunity" for consideration by *eLife*. Your article has been reviewed by three peer reviewers, and the evaluation has been overseen by a Reviewing Editor and Anna Akhmanova as the Senior Editor. The following individual involved in review of your submission has agreed to reveal their identity: S. Celeste Morley (Reviewer #1).

The reviewers have discussed the reviews with one another and the Reviewing Editor has drafted this decision to help you prepare a revised submission.

Your work identifies an important function of formin-like 1 (FMNL1) in activated T cell transendothelial migration (TEM).You developed a FMNL1 KO mouse and this mouse showed amazingly no immune defects (particularly in T cell differentiation and activation) during homeostasis. You, however, utilized two autoimmune disease models, type 1 diabetes and EAE, and showed impaired trafficking of activated FMNL1 KO T cells to the site of inflammation and also reduced disease incidence. You then investigated the mechanisms that caused impaired trafficking of FMNL1 KO T cells to nonlymphoid tissues. FMNL1 KO T cells have impaired diapedesis during TEM because FMNL1 mediates posterior perinuclear actin polymerization during TEM. This is important for transmigration of the large nucleus in lymphocytes across endothelial barriers, allowing for diapedesis through restrictive openings.

Revisions potentially requiring new data:

1) It would be helpful information in interpreting baseline changes in naïve and effector T cells if they could provide immunoblots of detergent soluble and insoluble fractions of F-actin and total cellular actin in the supplementary data, to see if it is altered by FMNL-1 deficiency. As they report fairly minor changes (but statistically significant, and likely physiologically significant changes) in the fold-change of F-actin increase following chemokine stimulation at 15 sec, it would be highly helpful to know if the baseline is beginning from the same place. (e.g. is there already 2x F-actin in WT cells? or half as much? Where are the FMNL-1 KO T cells starting from in terms of F-actin total content?)

2) The authors showed that there was no obvious defect when they stimulated polyclonal T cells from FMNL1 KO mice with anti-CD3 and anti-CD28 antibodies in vitro in Figure 2—figure supplement 1. However, I could not find evidences for FMNL1 KO OT-I and OT-II T cells activated in vitro with their cognate OVA peptides. Did they show similar TCR responses as control cells? This issue is important because experiments in Figure 1D were done by transferring these cells.

---

## [Author Response]

Revisions potentially requiring new data:1) It would be helpful information in interpreting baseline changes in naïve and effector T cells if they could provide immunoblots of detergent soluble and insoluble fractions of F-actin and total cellular actin in the supplementary data, to see if it is altered by FMNL-1 deficiency. As they report fairly minor changes (but statistically significant, and likely physiologically significant changes) in the fold-change of F-actin increase following chemokine stimulation at 15 sec, it would be highly helpful to know if the baseline is beginning from the same place. (e.g. is there already 2x F-actin in WT cells? or half as much? Where are the FMNL-1 KO T cells starting from in terms of F-actin total content?)

We agree that this is an important question. We have included data on the baseline levels of F-actin in unstimulated control and FMNL1 KO T cells (in Figure 6C). These data show similar baseline levels of F-actin in control and KO cells. The baseline F-actin data were obtained by Phalloidin staining and flow cytometry rather than biochemically as suggested. However, we believe that the addition of these baseline F-actin data in control and KO T cells should sufficiently address the reviewers’ question.

2) The authors showed that there was no obvious defect when they stimulated polyclonal T cells from FMNL1 KO mice with anti-CD3 and anti-CD28 antibodies in vitro in Figure 2—figure supplement 1. However, I could not find evidences for FMNL1 KO OT-I and OT-II T cells activated in vitro with their cognate OVA peptides. Did they show similar TCR responses as control cells? This issue is important because experiments in Figure 1D were done by transferring these cells.

We thank the reviewers for pointing this out and have added data showing that TCR transgenic FMNL1 KO T cells are in vitro activated and proliferate similarly to control T cells in response to cognate peptide (added as Figure 3—figure supplement 1). Specifically, we have included data from control and FMNL1 KO 2D2-TCR T cells, which also support that in the context of self-reactive T cells control and KO T cells are similarly activated in vitro. We have obtained similar results with OT-I and OT-II T cells but think that it would be redundant to add all these data in the paper.